# Decoding contextual influences on auditory perception from primary auditory cortex

**Bernhard Englitz[1,2]\*, Sahar Akram[3], Mounya Elhilali[4], Shihab Shamma[1,5]**

[1]Institute for Systems Research, University of Maryland, College Park, United States; [2]Computational Neuroscience Lab, Donders Institute for Brain Cognition and Behavior, Nijmegen, Netherlands; [3]Research Data Science, Meta Platforms, Menlo Park, United States; [4]Department of Electrical and Computer Engineering, Johns Hopkins University, Baltimore, United States; [5]Equipe Audition, Ecole Normale Supérieure, Paris, France

\*For correspondence: englitz@science.ru.nl

**Competing interest:** The authors declare that no competing interests exist.

## eLife Assessment

This **important** study explores the neural basis for a well known auditory illusion, often utilized in movie soundtracks, in which a sequence of two complex tones can be perceived as either rising or falling in pitch depending on the context in which they are presented. **Convincing** single-neuron data and analyses are presented to show that correlates of these pitch-direction changes are found in the ferret primary auditory cortex. While these findings provide an interesting link between cortical activity and perception, the manuscript could be clearer on the wider implications of the failure of traditional decoding models to account for these results.

**Abstract** Perception can be highly dependent on stimulus context, but whether and how sensory areas encode the context remains uncertain. We used an ambiguous auditory stimulus – a tritone pair – to investigate the neural activity associated with a preceding contextual stimulus that strongly influenced the tritone pair's perception: either as an ascending or a descending step in pitch. We recorded single-unit responses from a population of auditory cortical cells in awake ferrets listening to the tritone pairs preceded by the contextual stimulus. We find that the responses adapt locally to the contextual stimulus, consistent with human MEG recordings from the auditory cortex under the same conditions. Decoding the population responses demonstrates that cells responding to pitch-changes are able to predict well the context-sensitive percept of the tritone pairs. Conversely, decoding the individual pitch representations and taking their distance in the circular Shepard tone space predicts the *opposite* of the percept. The various percepts can be readily captured and explained by a neural model of cortical activity based on populations of adapting, pitch and pitch-direction cells, aligned with the neurophysiological responses. Together, these decoding and model results suggest that contextual influences on perception may well be already encoded at the level of the primary sensory cortices, reflecting basic neural response properties commonly found in these areas.

## Introduction

In real world scenarios, the elements of the sensory environment do not occur independently (*Lewicki, 2002*; *Smith and Lewicki, 2006*). Temporal, spatial and informational predictability exists within and across modalities, already as a consequence of the basic physical properties, such as spatial

and temporal continuity (*Rao and Ballard, 1999*). Neural systems make efficient use of this inherent predictability of the environment in the form of expectations (*Sohn et al., 2019*). Expectations are valuable, because they provide an internal mechanism to recognize stimuli faster (*Simon and Craft, 1989*) and more reliably under noisy conditions (*Lawrance et al., 2014*) with speech being of specific relevance for humans (*Norris et al., 2016*). As a consequence, the same stimulus can be perceived differently, depending on the context it occurs in, for example which stimuli it is preceded by or which stimuli co-occur with it (*Phillips et al., 2017*). We can thus study the expectation underlying a percept by studying the nature of the contextual influence.

Within audition, several forms of contextual influences have been found to shape perception. They range from spatial (e.g. localization in different contexts), to grouping (e.g. ABA sequences, *Bregman, 1994*) and phonetic (*Holt, 2005*) contextual influences. A striking example in human communication concerns the perception of certain syllable sequences, such as an ambiguous syllable between /ga/ and /da/ preceded by either /al/ or /ar/. In both, the second syllable is physically identical, but is heard as */da/* or */ga/* depending on the preceding syllable (*Lotto and Holt, 2006*). Subsequent psycho-acoustic investigations have revealed that this effect still occurs if the preceding syllable was replaced by appropriately chosen tone sequences, that it persists with substantial silent gaps between the two syllables, and that only very few tones are in fact necessary to bias the percepts one way or another (*Holt, 2005*). Hence, these contextual effects are likely not linguistic in nature, but reflect more basic adaptive neural mechanisms. Different interpretations have been provided to interpret these findings, such as the enhancement of contrasts (*Holt, 2005*; *Ulanovsky et al., 2003*).

Here, we investigate the neural correlates of these contextual effects using a simplified paradigm, in which the context reliably biases the percept of an *ambiguous* acoustic stimulus: A sequence of

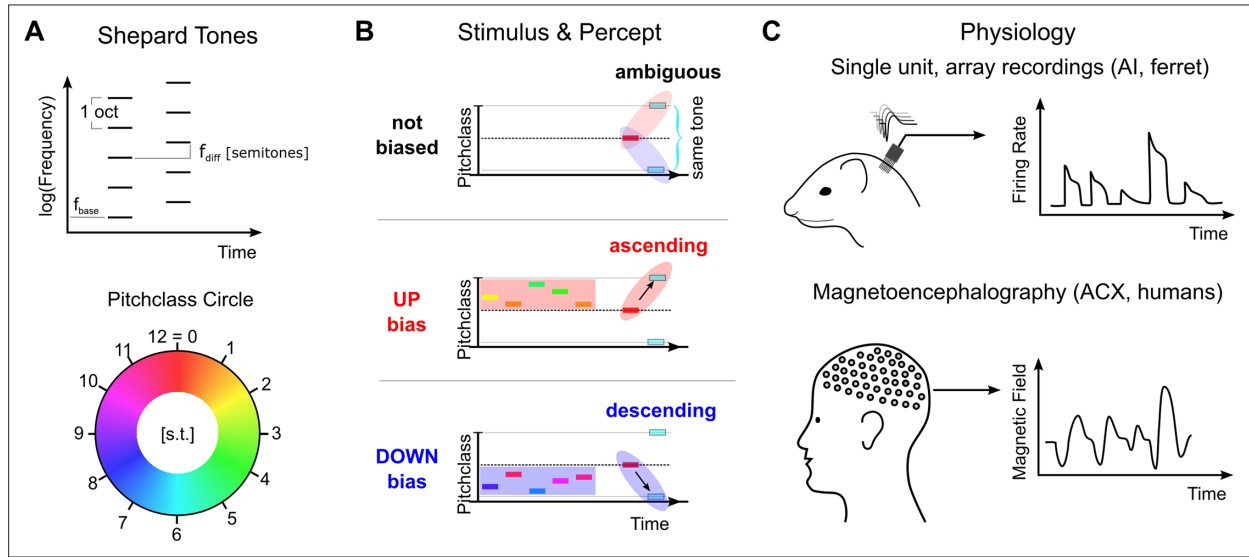

**Figure 1.** Stimulus design and recording techniques. (**A**) Shepard tones are acoustic complexes, composed of octave-spaced pure-tones (top). Each Shepard tone is uniquely characterized by its base frequency $f_{base}$, and the difference between two Shepard tones by the difference of their base frequencies $f_{diff}$, usually given in semitones. A Shepard tone shifted by a full octave 'projects' onto itself, and is therefore the physically same stimulus. The space of Shepard tones therefore forms a circle (bottom), in which each stimulus can be characterized as a so-called pitch class in semitones, which runs from 0 to 12, corresponding to a full octave. The base pitch class 0 was here chosen to correspond to $f_{base}$ = 440 Hz. We use the displayed color mapping (hue) throughout the paper. (**B**) We used the tritone paradox – a sequence of two Shepard tones – to investigate how ambiguous percepts are resolved, for example by preceding stimuli. In the tritone paradox, two Shepard tones are presented, which are separated by half an octave (6 semitones). Listeners, asked to judge the relative pitch between the two Shepard tones, are ambiguous as to their percept of an ascending or a descending step. If the ambiguous Shepard pair is preceded with a sequence of Shepard tones with pitch classes above the first but below the second tone (red area), listeners report an ascending percept. Conversely, if the ambiguous Shepard pair is preceded by a sequence of Shepard tones with pitch classes below the first, but above the second tone (blue area), listeners perceive a descending step. The neural representation of this contextual influence is not known, and we conducted a series of physiological and psychophysical experiments to elucidate the neural basis. (**C**) Neural responses from individual neurons were collected in awake ferrets from the auditory cortex (left). Individual neurons modulated their firing rate during the presentation of the stimulus sequence and exhibited tuned responses (see **Fig. S1**). Using MEG recordings, we also collected neural responses from populations of neurons in auditory cortex from human subjects, performing the up/down discrimination task. The amplitude of the magnetic field was modulated as a function of time during the stimulus presentation.

two *Shepard* tones (*Shepard, 1964*), differing by half an octave in the frequencies of its constituent tones, can be perceived as ascending or descending in pitch. Shepard tones are complex tones with octave spaced constituent tones (*Figure 1A*). This percept can be reliably manipulated by presenting a suitably chosen sequence of Shepard tones before, setting up different contexts (*Figure 1B*). This contextual influence is highly effective, rapidly established and can last for multiple seconds (*Repp, 1997*; *Chambers and Pressnitzer, 2014*). Importantly, the ability to determine the changes in pitch has relevance for a wide spectrum of real-world tasks, ranging from distinguishing an approaching from a departing vehicle to distinguishing different emotions in human communication (*Chebbi and Ben Jebara, 2018*; *Ethofer et al., 2006*).

We presented various Shepard tone sequences to awake ferrets and humans while simultaneously performing single-unit population recordings using chronically implanted electrode arrays in the left primary auditory cortex, and MEG (magnetoencephalographic) recordings, respectively. Exposure to the contextual sequence resulted in localized adaptation that faded over the time-course of ~1 s, consistent with stimulus specific adaptation (*Ulanovsky et al., 2003*) and with findings from a related human MEG study (*Chambers et al., 2017*). A straight-forward decoding approach demonstrates that the perceived pitch-change direction can be directly related to the contextually adapted activity of direction selective cells, that is cells that have a preference for sounds with ascending or descending frequency over time (*Brosch and Schreiner, 2000*; *Brosch et al., 1999*). Conversely, decoding the represented Shepard tone pitches and their respective differences, predicts a repulsive effect, opposite to the perceived direction of pitch change. These underlying neuronal adaptation dynamics are consistent with changes in neural activity in the auditory cortex estimated from human MEG recordings collected for the same sounds.

We can account for these effects in a simplified model of the cortical representation based on known properties of pitch-change selective cells, which matches both the results from the directional- and the distance-decoding analysis. Further, the model is consistent with multiple observed properties of the neural representation, including tuning changes and directional tuning of individual neurons as well as the build-up of the contextual effect in humans.

## Results

We collected neural recordings from 7 awake ferrets (662 responsive, tuned single units) and from 16 humans (MEG recordings) in response to sequences of Shepard tones (the 'Bias') followed by an ambiguous, semi-octave separated test pair (*Figure 1B*). The human participants performed a two-alternative forced choice task, selecting between hearing an ascending or descending step in the test pair, while ferrets listened passively. It has been previously shown (*Chambers and Pressnitzer, 2014*; *Chambers et al., 2017*) that the presence of the Bias reliably influences human perception, towards hearing a pitch step 'bridging' the location of the Bias, that is if the Bias is located above the first tone an ascending step is heard (*Figure 1B* middle), and a descending one, if it is below (*Figure 1B* bottom). The present study investigates the neural representation underlying these modified percepts.

In the following, we first show that the bias sequence induces a local adaptation in the neural population activity in both passive (ferret) and active (human) condition (*Figure 2*, see Discussion for more details). Next, we demonstrate the effect of this adaptation on the stimulus representation by decoding the neural population response. We find a repulsive influence of the Bias sequence on the pitch classes in the pair, that is the pitch class of each Shepard tone shifts away from the Bias. This increases their distance in pitch class along the perceived direction (*Figure 3*) and is thus not compatible with a simple, pitch (class)-distance based decoder as an explanation of the directionality percept (*Figure 4*). We then provide a simplified model of neuronal activity in auditory cortex that captures both the population representation and the adaptive changes in tuning of individual cells (*Figure 5*). Finally, based on this model, we provide an alternative explanation for the directionality percept of the ambiguous pair by showing that the adaptation pattern of directional cells predicts the percept (*Figure 6*).

For simplicity, the term *pitch* is used interchangeably with *pitch class* as only Shepard tones are considered in this study. Pitch (class) is here mainly used as a term to describe the neural responses to Shepard tones, as in previous literature on the topic, and the fact that Shepard tones are composite stimuli that lead to a pitch percept.

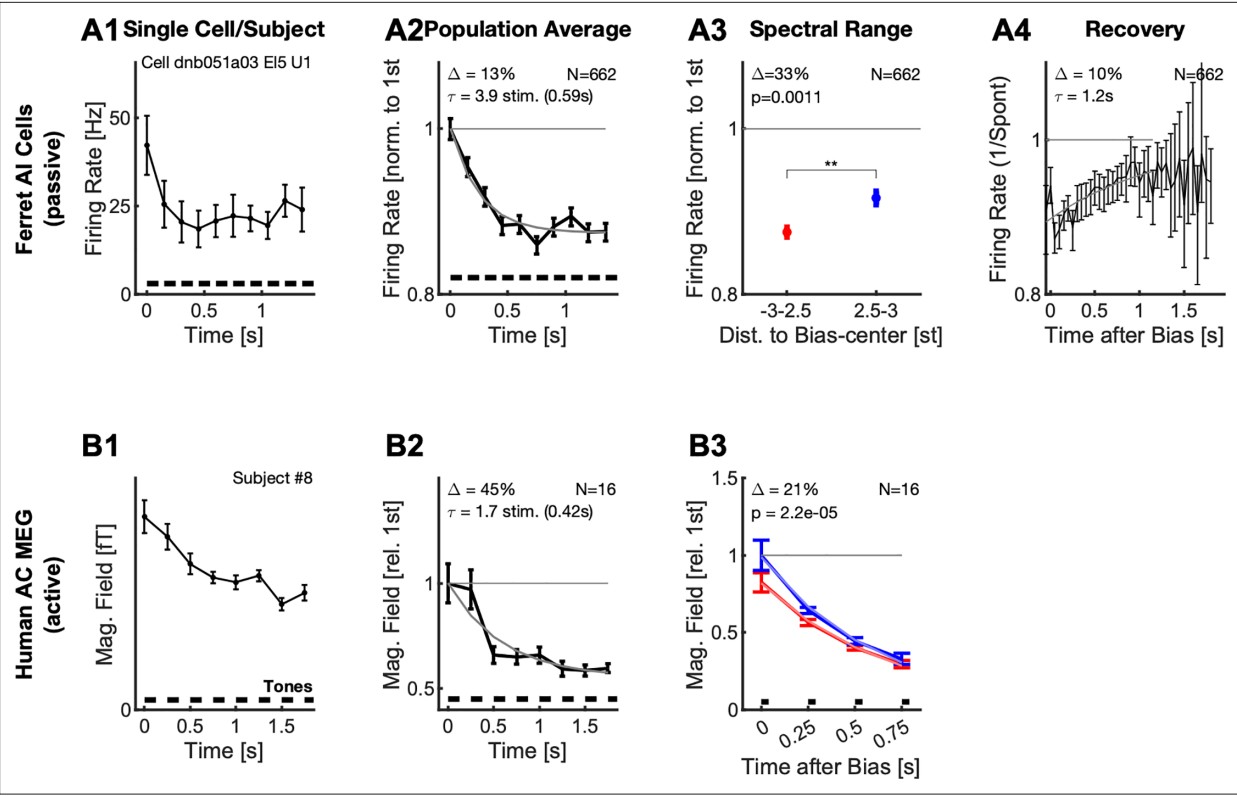

**Figure 2.** Neural responses adapt locally during the Bias sequence for awake ferrets (top) and behaving humans (bottom). (**A1**) During the presentation of the Bias sequence (10 tones, black bars), the neural response adapts over time (individual cell). This adaptation occurs for all parts of the response, shown here is the onset part (0–50ms, black). See *Figure 1* for more details on the different response types. Errorbars denote 2 SEM across trials. (**A2**) On the population level, the response reaches an adapted plateau 13% below the initial response after about 5 stimuli ($\tau$=3.9 stimuli, also for the onset response). This rate of reduction is similar to the rate of build-up of perceptual influence in human behavior (*Chambers and Pressnitzer, 2014*; *Chambers et al., 2017*). Errorbars denote 2 SEM across neurons. (**A3**) After the Bias the activity of the cells is significantly more reduced ($\Delta$=33%, p=0.001) around the center or the Bias (<2.5 semitones from the center) compared to the edges (2.5–3 semitones from the center). Errorbars denote 2 SEM across neurons. (**A4**) Recovery of spontaneous neural firing rates from the adaptation due to the Bias sequence progressed over multiple seconds with an exponential recovery time-constant of 1.2 s. (**B1**) In human auditory cortex the Bias sequences also evoked an adapting sequence of responses, here shown is the activity for a single subject (#8). Errorbars denote 2 SEM across trials. (**B2**) On average, the adaptation of the neural response proceeded with a similar time course as the single-unit response (**A2**), and plateaus after about 3–4 stimuli. Errorbars denote 2 SEM across subjects. (**B3**) Following the Bias, the activity state of the cortex is probed with a sequence of brief stimuli (35ms Shepard tones, after 0.5 s silence). Responses to probe tones in the same (red) semi-octave are significantly reduced (21% for the first time window, signed ranks test, p<0.0001) compared to the corresponding response in the opposite semi-octave (blue), indicating a local effect of adaptation. Errorbars denote one SEM across subjects.

The online version of this article includes the following figure supplement(s) for figure 2:

**Figure supplement 1.** Auditory cortex neurons respond tuned to Shepard tones.

## The contextual bias adapts the neural population locally

Adaptation to a stimulus is a ubiquitous phenomenon in neural systems (*Fairhall et al., 2001*; *Ulanovsky et al., 2004*; *Clifford et al., 2007*). Multiple kinds and roles of adaptation have been proposed, ranging from fatigue to adaptation in statistics (*Dean et al., 2005*; *Dean et al., 2008*; *Wen et al., 2012*; *Benucci et al., 2013*) to higher-order adaptation (*Shechter and Depireux, 2006*; *Rabinowitz et al., 2011*). Since adaptation has previously been implicated in affecting perception, for example in the tilt-after effect in vision (*Seriès et al., 2009*; *Kohn and Movshon, 2004*), we start out by characterizing the adaptation in neural response during and following the Bias under awake (ferrets, single units) and behaving (humans, MEG) conditions. The Bias was matched to the choice from a previous human study (*Chambers et al., 2017*, see *Figure 1B*), that is it consisted of a sequence of 5 or 10 Shepard tones with pitch classes randomly drawn from a range of 5 semitones, symmetrically arranged around a central pitch class, for example a Bias sequence centered at 3 semitones had

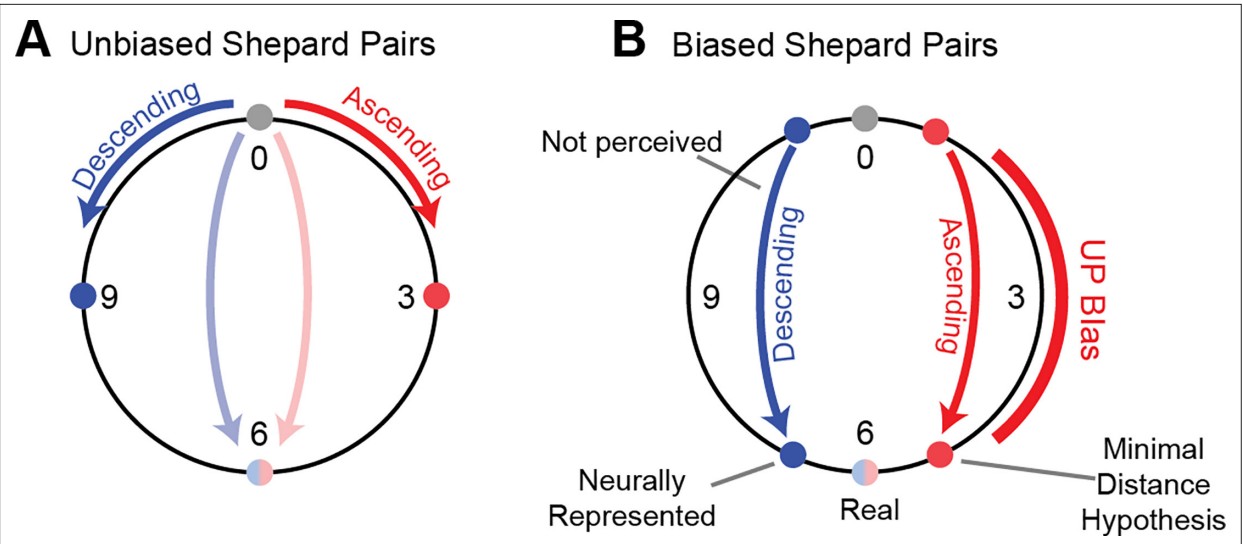

**Figure 3.** Repulsive shifts are not consistent with minimal distance hypothesis. (**A**) In the case of an unbiased Shepard pair, steps of less than 6 pitch classes lead to unambiguous percepts, e.g. a 0st to 3st steps leads to an ascending percept (red) and a 0st to 9st (i.e. 0st = >-3st) step to a descending percept (blue). Semi-octave (0st to 6st, blue/red) steps lead to an ambiguous percept, which strongly depends on the stimulus history (***Chambers et al., 2017***; ***Chambers and Pressnitzer, 2014***) and even the individual's specifics, such as language experience and vocal range (***Deutsch, 1986***). This suggests the *minimal distance hypothesis* which predicts the percept to follow the smaller of the two distances along the circle between the two pitch classes (**B**) In the case of an ambiguous Shepard pair (0st to 6st), preceded by a Bias sequence (red bar, right), here an *UP*-Bias, the ascending percept together with the *minimal distance hypothesis* would predict the distance between the Shepard tones to be reduced on the side of the *UP*-Bias (red dots). However, the population decoding shows that the distance between the tones is indeed increased on the side of the *UP*-Bias, challenging the *minimal distance hypothesis*.

individual tones drawn from 0.5 to 5.5 semitones. Relative to the ambiguous pairs, there were both an *up*- and a *down*-Bias, positioned above or below the first tone in the ambiguous pair, respectively.

In the single unit data, the average response strength decreases as a function of the position in the Bias sequence. Cells adapted their onset, sustained and offset response within a few tones in the biasing sequence (***Figure 2 A1***). This behavior was observable for the vast majority of cells (91%, p<0.05, Kruskal-Wallis test), and is thus conserved in the grand average (***Figure 2 A2***). The adaptation plateaued after about 3–4 stimuli (corresponding to a time-constant of 0.59 s) on a level about 13% below the initial level.

The single-unit response strength is reduced locally by the Bias sequence, that is more strongly for the range of Shepard tones occuring in the Bias. The responses to the tones in the ambiguous pair (***Figure 2 A3***, blue) – which are at the edges of the Bias sequence – are significantly less reduced (33%, p=0.0011, 2 group t-test) compared to within the bias (***Figure 2 A3***, blue vs. red), relative to the first responses of the bias. The average response was here compared against the unadapted response of the neurons measured via their Shepard tone tuning curve, collected prior to the Bias experiment (see Fig. S1 for some examples). This difference is enhanced if longer sequences are used and the entire non-biased region is measured (see below) The response strength remains adapted on the order of one second for single cells. The average spontaneous activity recovered with a time constant of 1.2 s (***Figure 2 A4***). The initial buildup before the reduction is probably due to offset responses of some cells.

For the human recordings, we obtained quite similar time-courses and qualitative response progressions. The neural response adapted both for individuals (***Figure 2 B1***) and on average (***Figure 2 B2***), with slightly faster time-constants (0.69 s), which could stem from the lower repetition rate (4 Hz compared to 7 Hz) used in the human experiments, potentially leading to less adaptation. To the contrary though, the amount of adaptation under behaving conditions in humans appears to be more substantial (40%) than for the average single units under awake conditions. While this difference could be partly explained by desynchronization which is typically associated with active behavior or attention (***Alishbayli et al., 2019***), general response adaptation to repeated stimuli is also typical in behaving humans (***Netser et al., 2011***). However, comparisons between the level of adaptation in

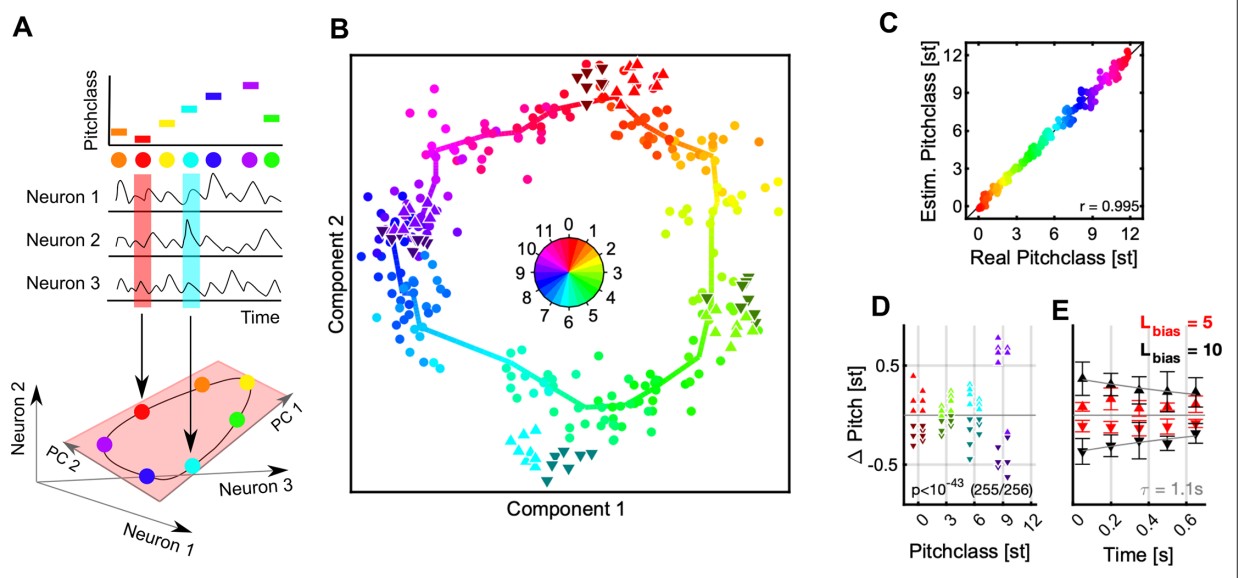

**Figure 4.** Population decoding predicts a Bias-induced, repulsive shift in pitch class. (**A**) We decoded the represented stimulus using dimensionality reduction techniques (see *Figure 2* for population-vector decoding). The stimulus identity (top) is reflected in the joint activity of all neurons (middle). If the neurons are considered as dimensions of a high-dimensional response space, the circular stimulus space of Shepard tones induces a circular manifold of responses, which lies in a lower dimensional space (light red plane). Colors represent a Shepard tone's pitch class, also in the following graphs. (**B**) The entire set of responses to the 240 distinct Shepard tones (from the various Bias sequences) is projected by the decoding into a low dimensional space (dots, hue = true pitch class), in which neighboring stimuli fall close to each other and the stimuli overall form a circle. The thick, colored line is computed from local averages along the range of pitch classes and emphasizes the circular structure. The Shepard tones in the ambiguous pairs are projected using the same decoder (denoted by the different triangles, hue = true pitch class), and roughly fall into their expected locations. However, if a stimulus was relatively above the Bias sequence (△, bright, upward triangles), their representation is shifted to higher pitch classes, compared to the same stimulus when located relatively below the Bias sequence (▽, dark, downward triangles). Hence, the preceding Bias repulsed the stimuli in the ambiguous pair in their represented pitch class. Both stimuli of the pair are treated equally here.(**C**) To demonstrate that decoding in this way is reliable, we compare real and estimated pitch classes (by taking the circular position in B) for each stimulus, which exhibits a reliable relation ($r$=0.995). (**D**) The influence of the Bias can be compared quantitatively by centering the represented test stimuli around their actual pitch class and inspecting the difference between the two different Bias conditions. After the Bias the decoded pitch class is shifted from their actual pitch class away from the biased pitch class range with high significance ($p<10^{-43}$, Wilcoxon-test). (**E**) The size of the shift is influenced by the length of the Bias sequence (5 tones = red, 10 tones = black) and the time between the Bias and the test tones ($\tau$=1.1 s). The errorbars indicate 1 SEM based on the number of number of test tones located above (N=24, upward triangles) and below (N=24, downward triangles).

The online version of this article includes the following figure supplement(s) for figure 4:

**Figure supplement 1.** Populationvector-based decoding also predicts a repulsive shift in pitch class.

MEG and single neuron firing rates may be misleading, due to the differences in the signal measured and subsequent processing.

Similarly to the neuronal data, in the MEG data the responses to probe tones in the same semi-octave (*Figure 2B3*, red) are significantly reduced (21%, signed ranks test, p<0.0001) compared to the corresponding response in the opposite semi-octave (blue). This local reduction is not surprising, given that single neurons in the auditory cortex can be well tuned to Shepard tones, with tuning widths of as little as 2–3 semitones (see *Figure 2—figure supplement 1E*). The detailed effect adaptation has on individual cells is studied in detail further below. Note, that the term local here could mean that a neuron is adapted in multiple octaves, but this is collapsed into the Shepard tone space.

In summary, both under awake (ferret) and behaving (humans) conditions, we find that neural responses adapt with similar time courses. The adaptation is local in nature, despite the global, that is wide-band, nature of the Shepard tones. Together, this suggests that adaptation may play an important role in explaining the effect the Bias sequence has on perceiving the ambiguous Shepard pair. Below we investigate the specific influence of the Bias on the detailed neural representation using the neuronal recordings from the ferret auditory cortex, which cannot be achieved currently with the human MEG data.

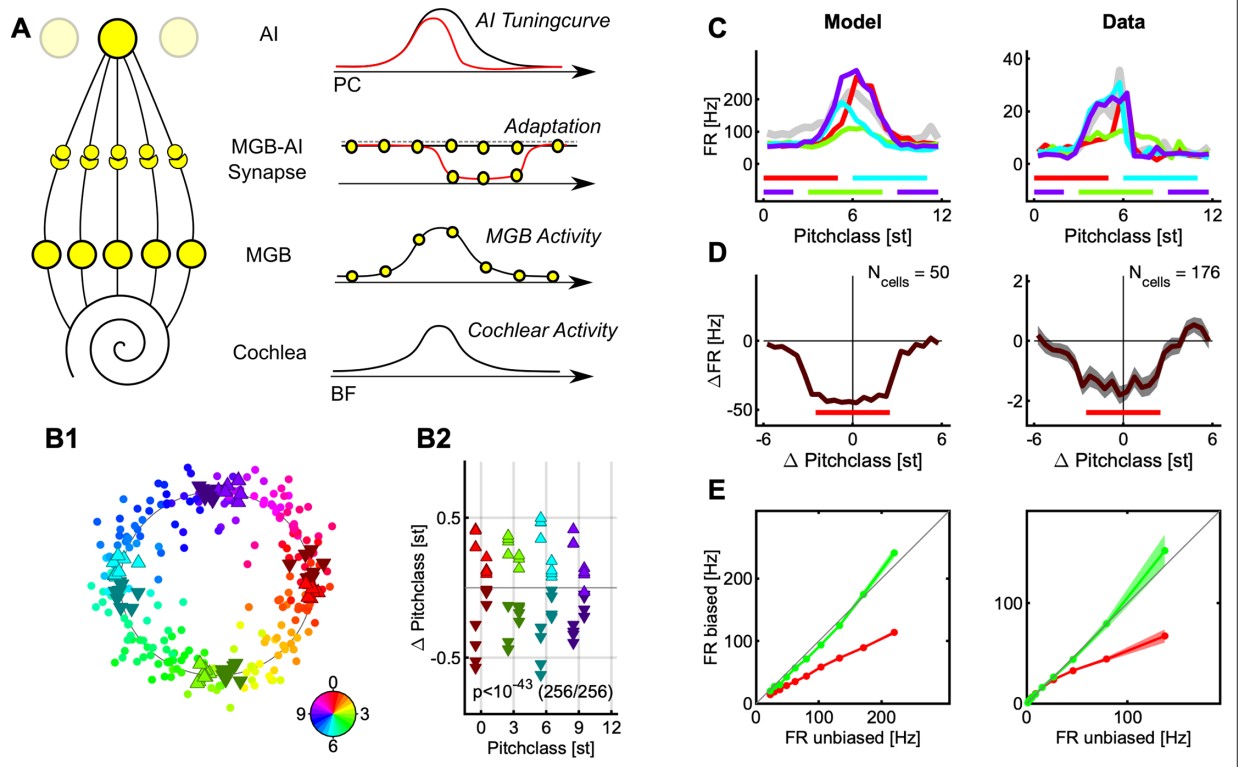

**Figure 5.** Distributed activity and local adaptation predict tuning changes and repulsive shifts. (**A**) We model the encoding by a simplified model, which starts from the cochlea, includes only one intermediate station (e.g. the MGB), and then projects to cortical neurons. The model is general in the sense that a cascaded version would lead to the same response, as long as similar mechanisms act on each level. A stimulus elicits an activity distribution along the cochlea (bottom) which is retained in shape on the intermediate level (2nd from below). In the native state, the stimulus is transferred to the cortical level without adaptation (2nd from top, black) and integrated by the cortical neuron (top, black). After a stimulus presentation, an adapted trough is left behind in the connections leading up to the cortical level (2nd from top, red), which reduces the cortical tuning curve locally. Since tuning curves closer to the center adapt more strongly, the stimulus representation in the neural population shifts away from the region of adaptation. (**B**) Applying the same analysis as above (*Figure 3*) for the real data leads again to a circular decoding (**B1**), with the estimated pitch classes of the tones the Shepard pair shifted repulsively by the preceding Bias (B2, for more details see the description of *Figure 4*). (**C**) Single cells show adaptation of their responses colocalized (different colors) with the biased region (colored bars, bottom). The Bias was presented in four different regions in separate trials, and the tuning of the cell probed in between the biasing stimuli. The left side shows a model example and the right side a representative, neural example. (**D**) Centered on the Bias, neurons in the auditory cortex adapt their response colocal with the Bias. The curves represent the difference in response rate between the unadapted tuning and the adapted tuning, again for model cells on the left and actual data on the right. (**E**) The presence of the Bias reduces the firing rate relative to the initial discharge rate, by ~40% (red), while the rate stays the same or is slightly elevated outside of the Bias regions (green) (see *Figure 3* for the decoding results and two related, incompatible models, which demonstrate noteworthy subtleties of the decoding process). The errorbars indicate 1 SEM across the set of neurons (see D for Ns).

The online version of this article includes the following figure supplement(s) for figure 5:

**Figure supplement 1.** Local and global (postsynaptic) adaptation cannot explain both influences of the Bias on the tuning (**A1–C1**) and the represented stimuli (**A2–C2**).

## The contextual bias repels the ambiguous pair in pitch

Adaptation can have a variety of effects on the represented stimulus attributes (see *Seriès et al., 2009*): stimulus properties can be attracted, repelled or left unchanged depending on the kind of adaptation. In the present paradigm, one hypothesis to explain the percept would be that *the bias attracts subsequent tones, thus reducing the distance along this side of the pitch-circle*, for example an *UP*-Bias would reduce an ambiguous 6 semitone step to a non-ambiguous 5 semitone step (*Figure 3*). To test this hypothesis, we decoded the represented stimulus using various decoding techniques from the neural population.

In population decoding the goal is to estimate a mapping from neural activity to stimulus properties, which assigns a stimulus to the population response. In the present context, this amounts to predicting the pitch class for a given neural response. Several decoding techniques exist which apply

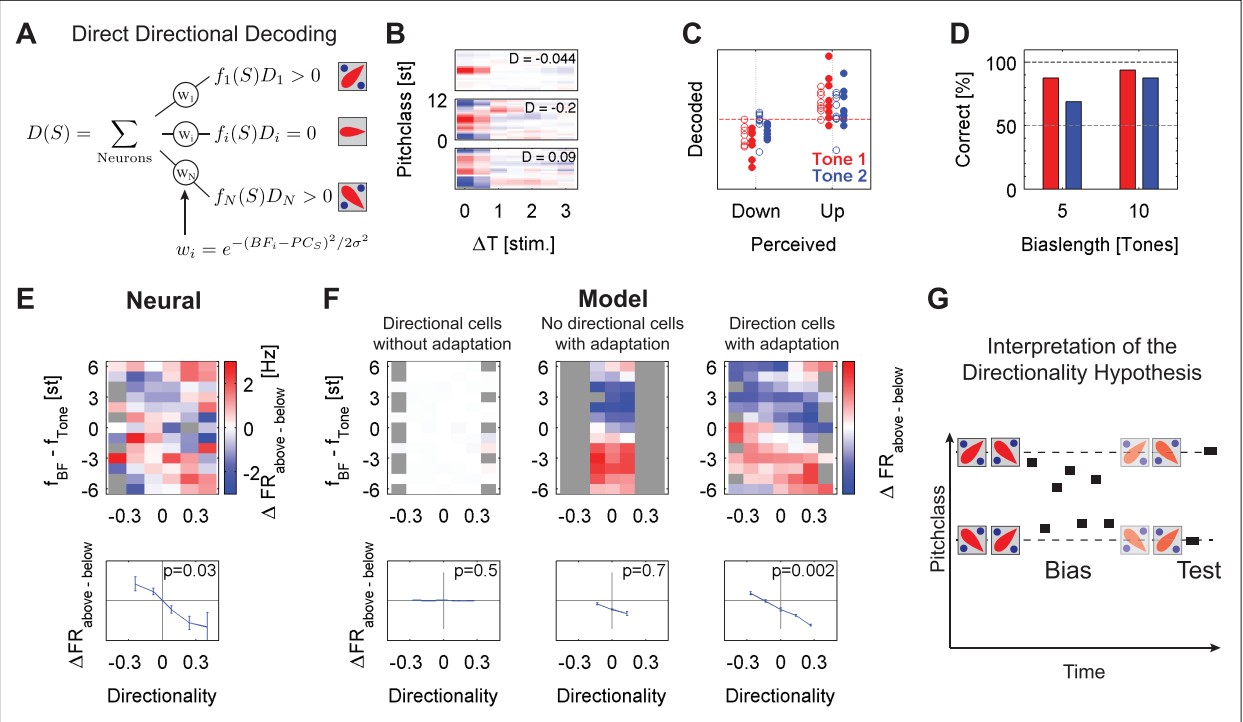

**Figure 6.** Decoding based on the directionality of the individual cells predicts the directional percept. (**A**) The predicted directional percept *D(S)* for a stimulus *S* is computed by the average over the cells activity $f_i(S)$, weighted by their directionality $D_i$ and their distance to the stimulus $w_i$. The inset images show prototypical *SSTRF*s of cells with the ascending (top, *up*), descending (bottom, *down*) and undirected (middle) directional preference. (**B**) Examples of three directional cells based on the Shepard-tone-based spectrotemporal receptive fields (*SSTRF*s). Directionality was determined by the asymmetry of the 2nd column of the *SSTRF* (response to previous stimulus), centered at the maximum (BF) of the first column (response to current stimulus, see Methods for details). As usual, time on the abscissa runs into the past. The middle cell for example is a down-cell, since it responds more strongly to a stimulus sequence 10st = >7 st, than 4st = >7 st based on the *SSTRF*. (**C**) The prediction of the decoding (ordinate) compared to the usually perceived direction for the two sequences (abscissa). Predictions depended on the length of the sequence (o=5 tones, •=10 tones) and the predicted tone (red = 1 st tone, blue = 2 nd tone). The dashed red line corresponds to a flat prediction. (**D**) Predictive performance increased as a function of Bias length and distance to the Bias, reflected as 1st (red) or 2nd (blue) tone after the Bias. Both dependencies are consistent with human performance and the build-up and recovery of adaptive processes. (**E**) The basis for the directional decoding can be analyzed by considering the entire set of Bias-induced differences in response, arranged by the directional preference of each cell (abscissa), and the location in BF relative to each stimulus in the Shepard pair (ordinate). Applying the analysis to the neural data, the obtained pattern of activity (top) is composed of two angled stripes of positive (red) and negative (blue) differential activity. For cells with BFs close to the pitch class of the test tones, the relative activities are significantly different (p=0.03, one-way ANOVA) between ascending and descending preferring cells, thus predicting the percept of these tones. Gray boxes indicate combinations of directionality and relative location which did not exist in the cell population. The errorbars indicate 1 SEM across the cells in each directionality bin. (**F**) Applied to a population of model neurons (as in *Figure 5*, see Methods for details) subjected to the same stimulus as the real neurons, in the absence of adaptation (left) no significant pattern emerges. If no directional cells are present (middle), adaptation leads to a distinct pattern for different relative spectral locations, but the lack of directional cells prevents a directional judgment. Finally, with adaptation and directional cells a pattern of differential activation is obtained, similar to the pattern in the neural data. T Cells located close to the target tone (near 0 on the ordinate) show a differential activity, predictive of the percept, which was used in the direct decoding above (shown separately in the lower plots). While these activities exhibit no significant dependence in the absence of adaptation or directional cells, the dependence becomes significantly characteristic with adaptation (p<0.001, one-way ANOVA, bottom right). (**G**) The above results can be summarized as a symmetric imbalance in the activities of directional cells after the Bias around it (right), which when decoded predict steps consistent with the percept, that is both are judged in their relative position to the Bias. Hence the percept of the pitch change direction is determined by the local activity, rather than by the circular distance between Shepard tones (*Figure 3*).

different algorithms and start from different assumptions. We here present a dimensionality reduction technique, based on principal component analysis, however, other techniques gave very similar results (e.g. Stochastic Neighborhood Embedding tSNE, *Maaten and Hinton, 2008*), or for a population vector decoding, see *Figure 4—figure supplement 1*.

Decoding techniques based on dimensionality reduction attempt to discover a new coordinate system, which accounts for a substantial portion of the variance within much fewer dimensions (*Figure 4A*). In other words, they estimate a new representation adapted to the intrinsic geometry

of the set of neural responses. In the case of Shepard tones, we predict this geometry to be circular (assuming the neural representation is not degenerate), given the circular nature of the Shepard tones (*Figure 1A*). As a circular variable can be represented (embedded) in a 2D Euclidean space, we only consider two dimensions of the decoding, typically the first two, if sorted by explained variance.

The projection of the neural data onto the first two principal components, forms indeed a circular arrangement (*Figure 4B*). Each point corresponds to a Shepard tone, with its actual pitch given by its color. The dimensionality reduction was based on the neural responses of 662 neurons to 240 distinct Shepard tones, compiled from the 32 biasing sequences (16 for both sequence lengths, 5 and 10), which covered the octave evenly. The orderly progression of colors indicates that a proximity in stimulus space leads to a proximity in neural response space.

To reassign a pitch class to each point, we estimated a continuous pitch-circle (*Figure 4B*, colorful polygon), by computing the local average of each set of 10 adjacent points (w.r.t. actual pitch) and interpolating in between these points. This decoder produces an excellent mapping between actual and estimated pitch classes on the training set ($r=0.995$, Pearson correlation, *Figure 4C*). Based on the responses to the Bias sequences, we have thus constructed a decoder of high accuracy.

Next, we apply the decoder to the responses of the Shepard tones in the ambiguous pair to estimate their represented pitch class, and check whether they are represented at their expected pitch class. We find that their pitch classes are shifted *away* from the Bias, that is tones in the pair that occur above the Bias are shifted further above (*Figure 4B/D/E*, △, bright, upward triangles), and vice versa (*Figure 4B/D/E*, ▼, dark, downward triangles). This result is highly significant ($p=10^{-41}$, exact ranks test, MATLAB *signrank*) and holds for all tested pitch classes in the pair ([0,3,6,9], *Figure 4D*). The size of the shift increases with the length of the Bias sequence (*Figure 4E*, red: L=5, black: L=10) and decreases with the temporal separation between Bias and ambiguous pair (*Figure 4E*, $\tau=1.1$ s). This time constant agrees well with the time course of recovery from adaptation (*Figure 2A4*). The effect size – ranging up to a total of 0.8 semitones – much greater than the human threshold of ~0.2 semitones for distinguishing two Shepard tones (internal pilot experiment, data not shown). Practically the same result is obtained using population vector decoding instead (*Figure 4—figure supplement 1*).

Consequently, the presence of the Bias has a *repulsive* effect on the tones in the pair. Therefore, their distance increases (when measured on the Shepard pitch circle) on the side of the pitch circle where the Bias was presented, for example for a 6 semitone step an *UP*-Bias leads to a represented 7 semitone (or correspondingly –5 semitone) step (*Figure 3B*). Hence, the population decoding suggests that a decoder based on circular distance in pitch *cannot* account for the effect of the Bias on the percept, since this would have predicted the distance to shrink on the side of the Bias.

## An SSA-like model accounts for repulsion and local adaptation

Before we can propose an alternative decoder for the directionality percept, we devise a basic neural model which is consistent with both the local nature of the adaptation (*Figure 2*) and the repulsion in pitch (*Figure 4*). This will serve to highlight a few boundary conditions coming from the neural activity and tuning properties that a complete model of the perceptual effect will have to obey. Generally, the model is a two-layer, tonotopically organized model with spectral integration for both layers and adaptation occurring in the transition from the first to the second layer (see *Figure 5A* for a depiction and Methods for more details).

As detailed below, the Bias not only leads to 'local' adaptation in the sense of 'cells close to the location of the Bias', but the adaptation even acts locally within the tuning curve of a given cell (*Figure 5D*, left). Hence, while global, postsynaptic adaptation (fatigue) has been shown to be sufficient in producing a repulsion from the adaptor in decoded stimuli (see *Figure 5—figure supplement 1*, *Seriès et al., 2009*; *Jin et al., 2005*), the local reduction of individual tuning curves actually observed in our data requires us to consider non-global adaptation in the model. Adaptation of this type is not unheard of in the auditory cortex, given that another cortical property of stimulus representation – stimulus specific adaptation (SSA, *Ulanovsky et al., 2003*; *Parras et al., 2017*) – is likely to rest on similar mechanisms.

The simplest possibility along these lines is very local adaptation, that is specific and limited to each stimulus. While this adaptation could account for the local changes in tuning-curves, it would not predict a repulsion to occur in decoding (*Figure 5—figure supplement 1A*, see Methods for details on the model implementation). Since this adaptation is assumed to be specific for each stimulus, the

population activity for this decoded stimulus would simply be scaled, which would leave the mean and all other moments the same (*Figure 5—figure supplement 1A*).

A more biological variant is adaptation that is based on the *internal*, *neural representation of the stimulus* (*Figure 5A*). As the auditory system has non-zero filter-bandwidths, every stimulus elicits a distributed, rather than a perfectly localized activity. At least up to the primary auditory cortex, acoustic stimuli are therefore represented in a distributed manner, in particular in the medial geniculate body (MGB) of the thalamus. If this distribution of activity adapts the corresponding channels locally, the tuning curves of cells in field AI of the primary auditory cortex – which receive forward input from the MGB – will be locally reduced, however, less local than in the point-like representation due to the width of the distribution in its inputs (*Figure 5—figure supplement 1C1*). On the other hand, decoding of pitches will be repulsive after an adaptor, because cells closer in BF to the adapting stimulus will integrate more adaptation (*Figure 5A* top right), and thus contribute less of their stimulus preference to the decoding. This imbalance shifts the average in the decoded pitch further away (*Figure 5—figure supplement 1C3*). For simplicity, the adaptation here is attributed to the incoming synaptic connections to AI, yet, it could equally be localized at an earlier or multiple levels.

The models discussed above (*Figure 5—figure supplement 1*) were implemented non-dynamically to illustrate the interaction between context and different types of adaptation. Based on the aforementioned considerations, we implemented a dynamical rate model including local adaptation and distributed representation (the latter two as in *Figure 5—figure supplement 1C*), which receives the identical stimulus sequences as were presented to the real neurons. The dynamical model provides a quantitative match to the adaptation of single cells and the repulsive representation of Shepard tones after the Bias and further allows us to estimate parameters of the underlying processing (*Figure 5B–E*).

First, when subjected to population decoding analysis as for the real data before, the model exhibits a very similar circular representation of the space of Shepard tones (*Figure 5B1*) and repulsive shifts in represented pitch class (*Figure 5B2*). The basis for this representation is analyzed in the following.

Local adaptation in the tuning of individual cells is retained in the model (*Figure 5C*). Individual cells showed adaptation patterns matched in location to the region where the Bias was presented (different colors represent different Bias regions), in comparison to the unbiased tuning curve (gray). Similarly, in the model, the locally implemented adaptation together with the distributed activity in the middle level leads to similarly adapted individual tuning curves.

To study the adaptation in a more standardized way, we computed the pointwise difference between adapted and unadapted tuning curve. These differences were then cocentered with the Bias before averaging (*Figure 5D*). Both for the onset (brown) and the sustained response (red), the reduction is highly local, with steep flanks at the boundaries of the Bias for both the model and the actual data. The offset data appears to show some local reduction but not as sharply defined as the other two.

In order to find the relative reduction, the response rates inside (red lines) and outside (green lines) the biased regions are plotted against the corresponding regions in the unbiased case (*Figure 5E*). While small firing rates appear to be less influenced, a reduction of ~40% stabilized for higher firing rates, largely independent of the response bin. This analysis can illustrate dependencies on firing rate and prevents degenerate divisions by small firing rates.

The adaptation encapsulated in the model makes only few assumptions, yet provides a qualitatively matched description of the neural behavior in response to the stimulus sequences (w.r.t. the present level of analysis). The model can now be extended to directional cells, to provide a novel explanation for the directionality percept of the biased Shepard pairs.

## The contextual Bias differentially adapts directional cells predicting the percept

The decoding techniques used in the previous sections relied only on the cells' tunings to the *currently* presented Shepard tone. However, cells in the auditory cortex also possess preferences for the *succession of two* (or more) stimuli, for example differing in frequency (*Brosch and Schreiner, 2000*; *Brosch et al., 1999*; *Brosch and Schreiner, 1997*). We hypothesized that perception of pitch steps could rely on the relative activities of cells preferring ascending and descending steps in frequency, or presently pitch class (*Figure 6A*), instead of the difference in decoded pitch class (as in the *minimal distance hypothesis,* disproven above).

We tested this *directional hypothesis* by decoding the perceived direction more directly by taking the directional preferences of each cell into account (*Figure 6A*). The directional percept is predicted by the population response weighted by each cell's directionality index (DI) and the distance to the currently presented stimulus (see Methods for details). The DI is computed from the cells' *SSTRF*s, which are estimated from the sequences of Shepard tones. This approach avoids estimating receptive fields from stimuli with different statistics. Directional cells have asymmetric spectro-temporal receptive fields (*SSTRF*, see *Figure 6B* for some examples): down-selective cells have *SSTRF*s with active zones (red) angled down (from past to future, current time being on the left), up-selective cells the opposite. In both cases, the *SSTRF* is dominated by the activity in response to the current stimulus, that is the recent past.

The directional decoding successfully predicts the percept for both stimuli in the ambiguous pair. Predictions are performed for each stimulus in the test pair separately. The step direction of the first stimulus is defined based on its relative position to the center of the Bias. For the analysis of the second Shepard tone in the pair, the step is assumed to be perceived on the side of the Bias, that is as previously shown in human perception (*Chambers and Pressnitzer, 2014*). Predictions (*Figure 6C*) for the first tone (red) were generally more reliable than for the second tone (blue), and predictions also improved for both tones with the length of the Bias sequence (5 tones = o, 10 tones = •). This dependence of prediction reliability is consistent with the certainty of judgment in human psychophysics (*Chambers and Pressnitzer, 2014*). On average, the prediction was correct in 88% and 95% for the first tone, for a Bias length of 5 and 10 tones, respectively, and 67% and 88% for the second tone, respectively (*Figure 6D*). In summary, the Bias seems to influence the relative activities of up- and down-preferring cells differentially above and below the Bias, such that responses from down-preferring cells prevail below the Bias, and up-preferring cells prevail above the Bias, predicting the human percept correctly.

We next investigate in what way the activities of directional cells are modified by the Bias to generate these perception-matched decodings. In the previous sections, we have seen that rather local adaptation occurs during the Bias and modifies the response properties of a cell. How does this adaptation affect a cell that has a directional-preference? To address this question, we distinguish the differential response between the *Up* and *Down* Bias as a function of a cell's directionality and pitch class separation from the test tone, that is Shepard tones in the ambiguous pair. Hence, the analysis (*Figure 6E/F*) plots the difference in response between a preceding *Up*- and *Down*-Bias (specifically: Bias-is-locally-above-tone minus Bias-is-locally-below-tone, color scale), as a function of the cells directional selectivity (abscissa) and the cell's best pitch class location with respect to each tone (ordinate). For the present analysis, the responses to the first and second tone in the pair were analyzed together. For the second tone, a cell thus contributes also to a second relative-to-tone bin (ordinate) at the same directionality, however, with a different set of responses. Also, each cell contributed for each tone in multiple locations, since multiple target tones (4) were tested in the paradigm.

For the neural data, the differential responses exhibit an angled stripe pattern, formed by a positive and a negative stripe (*Figure 6E* top). The stripes are connected at the top and bottom ends, due to the circularity of the Shepard space. The pattern of differential responses conforms to the *directional hypothesis*, if *down*-cells (left half) are more active than *up*-cells (right half) close to the pair tones (*Figure 6E*, ordinate around 0). This central region was considered here, since these are the cells that will respond most strongly to the tone. For the neural data, this differential activity is significantly dependent on the directionality of the cells (*Figure 6E* bottom, ANOVA, p<0.005).

## Extending the neuronal model to directionally sensitive cells

In order to better understand the mechanisms shaping this biasing pattern, the same analysis was applied to neural models including different properties (*Figure 6F*). The same model as in the previous section was used, with the only difference that the tuning of the cells extended in time to include two stimuli instead of only one, comparable to the *SSTRF*s of the actual cells (*Figure 6B*). To illustrate the effect of adaptation, three models are compared: without adaptation (*Figure 6F* left), without directionally tuned cells (*Figure 6F* middle), and with adaptation and directionality tuned cells (*Figure 6F* right).

Without adaptation, the cells do not show a differential response (*Figure 6F* left), since the Bias does not affect the responses in the test pair (Note, that there is a 200ms pause between the end of

the Bias and the test pair, such that the directionality itself cannot explain the pattern of responses). Here, the difference in activity around the current tone is not significant (ANOVA, p=0.5; *Figure 6F* left bottom).

Without directional cells, the pattern reflects only the difference in activity generated by the interaction of the Bias with the adaptation. The lack of directional cells limits the pattern to a small range of directionalities, generated by estimation inaccuracy. Hence, the local pattern of differential response around the test tone is not significantly modulated, due to the lack of directional cells to span the range (ANOVA, p=0.7; *Figure 6F* middle bottom).

In the model with adapting and directional cells, the pattern resembles the angled double stripe pattern from the neural data. The stripes in the pattern are generated by the adaptation, whereas the directionality of the cells leads to the angle of those stripes. Locally around the test tone, this difference shows a statistically significant dependence on the directionality of the cells (ANOVA, p<0.0001; *Figure 6F* right bottom).

The resulting distribution of activities in their relation to the Bias is, hence, symmetric around the Bias (*Figure 6G*). Without prior stimulation, the population of cells is unadapted and thus exhibits balanced activity in response to a stimulus. After a sequence of stimuli, the population is partially adapted (*Figure 6G* right), such that a subsequent stimulus now elicits an imbalanced activity. Translated concretely to the present paradigm, the Bias will locally adapt cells. The degree of adaptation will be stronger, if their tuning curve overlaps more with the biased region. Adaptation in this region should therefore most strongly influence a cell's response. For example, if one considers two directional cells, an *up*- and a *down*-selective cell, cocentered in the same spectral location below the Bias, then the Bias will more strongly adapt the *up*-cell, which has its dominant, recent part of the *SSTRF* more inside the region of the Bias (*Figure 6G* right). Consistent with the percept, this imbalance predicts the tone to be perceived as a descending step relative to the Bias. Conversely, for the second stimulus in the pair, located above the Bias, the *down*-selective cells will be more adapted, thus predicting an ascending step relative to the previous tone.

*In summary*, taking into account the directional selectivities of the population of cells and local neural adaptation, the changes in the directional percept induced by the Bias sequences can be predicted from the neural data. Specifically, the local adaptation of specifically the directionally selective cells caused by the Bias underlies the imbalance in their responses, and thus is likely to be the underlying mechanism of the biase Shepard tones percept.

## Discussion

We have investigated the physiological basis underlying the influence of stimulus history on the perception of pitch-direction, using a bistable acoustic stimulus, pairs of Shepard tones. Stimulus history is found to persist as spectrally localized adaptation in animal and human recordings, which specifically shapes the activity of direction-selective cells in agreement with the percept. The adaptation's spectral and temporal properties suggest a common origin with previously described mechanisms, such as stimulus specific adaptation (SSA). Conversely, the typically assumed (*Shepard, 1964*; *Repp, 1997*), but rarely explicitly discussed, circle-distance hypothesis in Shepard tone judgements is in conflict with the repulsive effect on cortically represented pitch revealed presently using different types of population decoding. While our entire study was based on Shepard tones, we hypothesize that the underlying mechanisms will influence many other stimuli as well, although the perceptual salience will depend on the level of ambiguity.

### Relation to previous studies of Shepard tone percepts and their underlying physiology

While context-dependent auditory perception of Shepard tones has been studied previously in humans, we here provide a first account of the underlying neurophysiological representation. Previous studies have considered how the stimulus context influences various judgements, for example whether subsequent tones influence each other in frequency (*Ronken, 1972*; *Raviv et al., 2012*), whether a sound is continuous (*Riecke et al., 2011*) or which of two related phonemes is perceived (*Holt, 2005*). In the present study, we chose directional judgements, due to the fundamental role frequency-modulations play in the perception of natural stimuli and language. We find the preceding stimulus to locally bias

directional cells, such that on a population level the first Shepard tone is perceived as a step downward, and the second tone as a step upward for an UP Bias, and conversely for a DOWN Bias. While the present study cannot directly rule out that the local adaptation occurs before the thalamo-cortical junction, both physiology (*Wehr and Zador, 2005*) and psychophysical (binaural fusion, *Deutsch, 1992*) results suggest a location beyond the olivary nuclei.

The success of the directional decoder in linking the cellular activity in A1 to the human percept under the same stimulus conditions is of remarkable accuracy, ~90%. We would like to emphasize that the use of the directional decoder is equally plausible as the use of a preferred frequency based decoder. In both cases, it is assumed that a downstream region in the brain pools the neural responses from A1 assuming either only a frequency preference, or a combination of frequency and direction preference. The elegance of the directional decoder is that it makes a direction connection to well-known directional response characteristics of auditory cortical neurons (see below), while avoiding any mechanisms specific to Shepard tones, such as the computation of the circular distance.

Are these results compatible with the cellular mechanisms that give rise to direction selectivity? The cellular mechanisms underlying the emergence of directional selectivity in the auditory system have been elucidated in recent years using in vivo intracellular recordings (*Ye et al., 2010*; *Zhang et al., 2003*; *Kuo and Wu, 2012*). Two mechanisms have been identified in the auditory cortex, (i) excitatory inputs with different timing and spectral location and (ii) excitatory and inhibitory inputs with different spectral location. In both mechanisms local adaptation by prior stimulation would tend to equalize direction selectivity, by diminishing (i) one excitatory channel or (ii) either the inhibitory or the excitatory channel. The observed changes in response properties under local stimulation are thus compatible with the network mechanisms underlying direction selectivity. This makes the prediction that the presentation of FM-sweeps of one direction should bias subsequent perception to the opposite direction. The timescales tested in these studies (*Ye et al., 2010*; *Shamma et al., 1993*) are similar to what was presently used, for example *Ye et al., 2010* used FM sweeps that lasted for up to 200ms, which is quite comparable to our SOA of 150ms. Psychophysical evidence in this respect has been observed previously in terms of threshold shifts of directional perception, which are in agreement with a local bias influencing the directional percept of subsequent stimuli (*Gardner and Wilson, 1979*; *Dawe et al., 1998*). More specific adaptation paradigms are required to resolve some of the more detailed effects for example local differences across the octave (*Dawe et al., 1998*).

Conversely, we could disprove the hypothesis that directional judgements are based on the distance between the tones on the circular Shepard space. Earlier studies on directional judgements of Shepard pairs have – implicitly or explicitly – used the circular nature of the Shepard space to predict the percept (*Repp, 1997*; *Dawe et al., 1998*; *Deutsch, 1986*), starting from the fundamental work of *Shepard, 1964* The original idea was to construct a stimulus with tonal character but ambiguous pitch, and as such it has interesting applications in the study of pitch perception. However, as presently shown, the percept of directionality does not rest on the circular construction. This conclusion is obtained by decoding the represented pitch of the Shepard tones in the context of different biasing sequences. This analysis demonstrated that the biasing sequence exerts a repulsive rather than an attractive effect on the pitch of following stimuli. Repulsive effects of this kind have been widely investigated in the visual literature, in particular the *tilt after-effect* (*Seriès et al., 2009*; *Kohn and Movshon, 2004*; *Jin et al., 2005*; *Gibson and Radner, 1937*; *Kohn, 2007*), where exposure to a single oriented grating perceptually repels subsequently presented gratings of similar orientation. Repulsive effects have also been described in the auditory perception (*Holt, 2005*; *Ronken, 1972*; *Riecke et al., 2011*), but not in auditory physiology. In conclusion, we find the percept to be inconsistent with the increase in the circular distance in the Shepard tone space.

An interesting approach would be to provide a Bayesian interpretation for the effect of the Bias on the cortical representation. Typically an increase in activity is considered as a representation of the prior occurrence probability of stimuli (*Huys et al., 2007*). Given the local reduction in activity described above, this interpretation would, however, not predict the percept. Alternatively, one could propose to interpret the *negative* deviation, that is local adaptation, as the local magnitude of the prior, which could be consistently interpreted with the percept in this paradigm, as has been proposed before (*Chambers et al., 2017*). Recordings from different areas in the auditory cortex might, however, show different characteristics, including a sign inversion.

## Relation to other principles of adaptation in audition

Adaptation has been attributed with several functions in sensory processing, ranging from fatigue (adaptation in excitability of spiking), representation of stimulus statistics (*Fairhall et al., 2001*), compensation for stimulus statistics (*Benucci et al., 2013*), sensitization for novel stimuli (*Ulanovsky et al., 2003*; *Pérez-González et al., 2005*) and sensory memory (*Ulanovsky et al., 2004*; *Jääskeläinen et al., 2007*). Adaptation is also present on multiple time-scales, ranging from milliseconds to minutes (*Fairhall et al., 2001*; *Ulanovsky et al., 2004*; *Shechter and Depireux, 2006*; *Gourévitch et al., 2009*). Based on the time-scales of the stimulus and the task-design, the present experiments mainly revealed adaptation in the range of fractions of a second. Adaptation can be global – in the sense that a neuron responds less to all stimuli – or local – in the sense that adaptation is specific to certain, usually the previously presented stimuli, as in SSA (*Ulanovsky et al., 2003*; *Condon and Weinberger, 1991*). Here, adaptation was well confined to the set of stimuli presented before. Hence, the adaptation identified presently is temporally and spectrally well matched to SSA described before. In recent years, the research on SSA has focussed on the aspect of stimulus novelty (*Nelken and Ulanovsky, 2007*; *von der Behrens et al., 2009*; *Bäuerle et al., 2011*), as a potential single-cell correlate of mismatch-negativity (MMN) recorded in human EEG and MEG tasks. While the connection between SSA and MMN appears convincing when it comes to some properties, for example stimulus frequency, it appears to not transfer in a similar way to other, still primary properties, such as stimulus level or duration, which elicit robust MMN (*Farley et al., 2010*). The present results reemphasize another putative role of SSA, namely sensory memory. Naturally, adaptation – if it is local – constitutes a 'negative afterimage' of the preceding stimulus history. Recent studies in humans suggest a functional role for this adapted state in representing properties of the task. This was recently demonstrated in an auditory delayed match-to-sample task, where a frequency-specific reduction in activity was maintained between the sample and the match (*Linke et al., 2011*), see also *Rinne et al., 2005*. Localized adaptation as described presently provides a likely substrate for such a sensory memory trace.

## Future directions

While in human perception task engagement is not necessary to be influenced by the biasing sequence, a natural continuation of the present work would be to record from behaving animals. This would allow us to investigate potential differences in neural activity depending on the activity state, and how individual neurons contribute to the decision on a trial-by-trial basis (*Haefner et al., 2016*; *Haefner et al., 2013*). Furthermore, the current study was limited to the primary auditory cortex of the ferret, but secondary areas as well as parietal and frontal areas could also be involved and should be explored in subsequent research. Switching to mice as an experimental species would allow us to differentiate the roles of different cell types better (*Natan et al., 2017*). On the paradigm level, an extension of the time between the end of the Bias sequence and the test pair would be of particular interest in the active condition, where human research suggests that the Bias can persist for more extended times than suggested by the decay properties of the adaptation in the present data set.

## Methods
### Experimental procedures

We collected single unit recordings from 7 adult, female ferrets (age: 6–12 months, *Mustela putorius furo*) in the awake condition. The animal experiments were performed in strict accordance with the recommendations in the Guide for the Care and Use of Laboratory Animals of the National Institutes of Health. All of the animals were handled according to approved institutional animal care and use committee (IACUC) protocols (Neurobiology of Auditory Cognition in Ferrets R-APR-23–24) of the University of Maryland, College Park.

We collected MEG recordings from 16 human subjects (9 female, average age: 26y, range 23–34 y) and psychophysical recordings from 10 human subjects (5 female, average age: 28y, range 25–32 y). All human experiments were performed in accordance with the ethical guidelines of the University of Maryland. The human experiments were approved by the Institutional Review Board of the University of Maryland, College Park under the project number 1467378 ("The Adaptive Auditory Mind"). Listeners read and signed a consent form regarding data use and processing before data collection.

## Surgical procedures

A dental cement cap and a headpost were surgically implanted on the animal's head using sterile procedures, as described previously (*Fritz et al., 2003*). Microelectrode arrays (Microprobes Inc, 32–96 channels, 2.5 MOhm, shaft ø=125 μm, various planar layouts with 0.5 mm interelectrode spacing) were surgically implanted in the primary auditory cortex AI at a depth of ~500 μm, for two animals sequentially on both hemispheres. A custom-designed, chronic drive system (see https://code.google.com/archive/p/edds-array-drive/) was used in some recordings to change the depth of the electrode array.

## Physiology : stimulation and recording

Acoustic stimuli were generated at 80 kHz using custom written software (available in the data repository *Englitz et al., 2024*, see below) in MATLAB (The Mathworks, Natick, USA) and presented via a flat calibrated (within +/-5 dB in the range 0.1–32 kHz using the inverse impulse response) sound system (amplifier: Crown D75A; speaker: Manger, flat within 0.08–35 kHz). Animals were head-restrained in a standard position in a tube inside a soundproof chamber (mac3, Industrial Acoustics Corporation). The speaker was positioned centrally above the animal's head and calibration was performed for the animal head's position during recordings.

Signals were pre-amplified directly on the head (1 x or 2 x, Blackrock/TBSI) and further amplified (1000 x, Plexon Inc) and bandpass-filtered (0.1–8000 Hz, Plexon Inc) before digitization ([–5,5]V, 16 bits, 25 kHz, M-series cards, National Instruments) and storage/display using an open-source DAQ system (*Englitz et al., 2013*). Single units were identified using custom written software for spike sorting (for details see *Englitz et al., 2009*). All subsequent analyses were performed in Matlab.

## Magnetoencephalography and psychophysics : stimulation, recording, and data analysis

Acoustic stimuli were generated at 44.1 kHz using custom written software in MATLAB and presented via a flat calibrated (within +/-5 dB in the range 40–3000 Hz) sound system. During MEG experiments, the sound was delivered to the ear via sound tubing (ER-3A, Etymotic), inserted with foam plugs (ER-3–14) into the ear canal, while during psychophysical experiments an over-the-ear headphone (Sony MDR-V700) was used. While the limited calibration range (due to the sound tubing) is not optimal, it still encompasses >6 octaves/constituent tones for every Shepard tone. Sound stimuli were presented at 70 dBSPL. Magnetoencephalographic (MEG) signals were recorded in a magnetically shielded room (Yokogawa Corp.) using a 160 channel, whole-head system (Kanazawa Institute of Technology, Kanazawa, Japan), with the detection coils (ø=15.5 mm) arranged uniformly (~25 mm center-to-center spacing) around the top part of the head. Sensors are configured as first-order axial gradiometers with a baseline of 50 mm, with field sensitivities of >5 fT/Hz in the white noise region. Three of the 160 channels were used as reference channels in noise-filtering methods (*de Cheveigné and Simon, 2007*). The magnetic signals were band-passed between 1 Hz and 200 Hz, notch filtered at 60 Hz, and sampled at 1 kHz. Finally, the power spectrum was computed and the amplitude at the target rate of 4 Hz was extracted (as in *Elhilali et al., 2009*, all magnetic field amplitudes in *Figure 2B* represent this measure).

Subjects had to press one of two buttons (controller held in the right hand, away from the sensors) to indicate an ascending or a descending percept. Subjects listened to 120 stimuli in a block, and completed 3 blocks in a session, lasting ~1 hr.

## Acoustic stimuli

All stimuli were composed of sequences of Shepard tones. A Shepard tone is a complex tone built as the sum of octave-spaced pure-tones. To stimulate a wide range of neurons, we used a flat envelope, that is all constituent tones had the same amplitude. Phases of the constituents tones were randomized for each trial, to prevent any single, fixed phase relationship from influencing the pitch percept. Each Shepard tone was gated with 5ms sinusoidal ramps at the beginning and end.

A Shepard tone can be characterized by its position in an octave, termed pitch class (in units of semitones), w.r.t. a base-tone. In the present study, the Shepard tone based on 440 Hz was assigned pitch class 0. The Shepard tone with pitch class 1 is one semitone higher than pitch class 0 and pitch class 12 is identical to pitch class 0, since all constituent tones are shifted by an octave and range

from inaudibly low to inaudibly high frequencies. Hence, the space of Shepard tones is circular (see *Figure 1B*). Across the entire set of experiments the duration of the Shepard tones was 0.1 s (neural recordings) / 0.125 s (MEG recordings) and the amplitude 70 dB SPL (at the ear).

We used two different stimulus sequences to probe the neural representation of the ambiguous Shepard pairs and their spectral and temporal tuning properties, (i) the Biased Shepard Pair and (ii) the Biased Shepard Tuning:

## Biased shepard pair

In this paradigm, an ambiguous Shepard pair (6 st separation) preceded by a longer sequence of Shepard tones, the Bias (see *Figure 1C*). The Bias consists of a sequence of Shepard tones (lengths: 5 and 10 stimuli) which are within 6 semitones above or below the first Shepard tone in the pair. These biases are called 'up' and 'down' Bias respectively, as they bias the perception of the ambiguous pair to be 'ascending' or 'descending', respectively, in pitch (*Chambers and Pressnitzer, 2014*; *Chambers et al., 2017*). A pause of different length ([0.05,0.2,0.5] s) was inserted between the Bias and the pair, to study the temporal aspects of the neural representation. Altogether we presented 32 different Bias sequences 4 base pitch classes ([0,3,6,9] st), 2 randomization (pitch classes and position in sequence), 2 Bias lengths (*Simon and Craft, 1989*; *Holt, 2005* stimuli, 'up' and 'down Bias), which in total contained 240 distinct Shepard tones. Their individual pitch classes in the Bias were drawn randomly and continuously from their respective range. Each stimulus was repeated 10 times. In all subsequent analyses, neural responses are averages over these repetitions, and all analyses are performed on the pooled data from all animals. For the neural data, these 240 different Shepard tones were also used to obtain a 'Shepard tone tuning' for individual cells (see *Figure 2—figure supplement 1*). The stimulus described above was presented to both animals and humans. The human psychophysical data were only used to reproduce the previous findings by *Chambers and Pressnitzer, 2014* with the current parameters. For the MEG recordings, a variation of the biased Shepard pair stimulus was used, which enabled the separate measurement of the activation state in the biased and the unbiased spectral regions. For this purpose a second sequence of Shepard tones (tone duration: 30ms; SOA: 250ms; pitch classes: 3 st above or below the tone of the pair) was inserted between the Bias sequence and the Shepard pair, with the time between the two adapted to include the duration of the sequence (2 s) and a pause after the Bias sequence ([0.5,1,2] s).

## Biased shepard tuning

For estimating the changes in the tuning curve of individual neurons, much longer sequences (154 Shepard tones) were presented to a subset of the neurons. The duration and stimulus onset asynchrony was matched to the Bias sequence. The Shepard tones in these sequences were chosen to maintain the influence of the Bias over the entire sequence, while intermittently probing the entire octave of semitones to estimate the overall influence of the Bias on the tuning of neurons. For this purpose, 5/6 (~83%) of the tones in the sequence were randomly drawn from one of the four Bias regions ([0–5], [3-8],[6-11],[9-2]st), while the 6th tone was randomly drawn from the entire octave, discretized to 24 steps (reminiscent of the studies of *Dean et al., 2005*). The 6th tone could thus be used to measure each neurons 'Shepard tuning' at a resolution of 0.5 semitones, adapted to different Bias locations. To avoid onset effects, a lead-in sequence of 15 Bias tones preceded the first tuning estimation tone. Individual stimulus parameters (intensity, durations of tone and interstimulus interval) were chosen as above. Five pseudorandom sequences were presented for each of the four Bias regions, repeated 6 or more times, providing at least 30 repetitions for each location in the tuning curve (Results of these conditions are shown in *Figure 5*). A randomly varied pause of ~5 s separated the trials.

### Unit selection

Overall, we recorded from 1467 neurons across all ferrets, out of which 662 were selected for the decoding analysis based on their driven firing rate (i.e. whether they responded significantly to auditory stimulation) and whether they showed a differential response to different Shepard tones. The thresholds for auditory response and tuning to Shepard tones were not very critical: setting the threshold low led to qualitatively the same result, however, with more noise. Setting the thresholds very high, reduced the set of cells included in the analysis, and eventually made the results less stable, as the cells did not cover the entire range of preferences to Shepard tones.

## Response type analysis

Whether a cell was adapting or facilitating within the Bias sequence was assessed by averaging the responses across all Bias sequences for a given cell separately. The resulting PSTH was then split up into Onset, Sustained and Offset bins, each 50ms in time, for each stimulus in the Bias. The sequence of response rates was then fitted with an exponential function, and the direction of the adaptation assessed by comparing the initial rate and the asymptotic rate. If the asymptotic rate exceeded the initial rate, a cell was classified as facilitating, conversely as adapting. The three response bins showed similar proportions of adapting response and were thus averaged to assign a single response type to a given cell, as reported in the Results.

## Population decoding

The represented stimuli in the ambiguous pair were estimated from the neural responses by training a decoder on the biasing sequences and then applying the decoder to the neural response of the pair. We used two different decoders to compare their results, one based on dimensionality reduction (PCA, Principal Component Analysis) and one based on a weighted population-vector, which both gave very similar results (see *Figure 4* and S2). For both decoders, we first built a matrix of responses which had the (240) different Shepard tones occuring in all Bias sequences running along one dimension and the neurons along the other dimension.

The PCA decoder performed a linear dimensionality reduction, utilizing the stimuli as examples and the neurons as dimensions of the representation. The data were projected to the first three dimensions, which represented the pitch class as well as the position in the sequence of stimuli (see *Figure 4A* for a schematic). As the position in the Bias sequence was not relevant for the subsequent pitch class decoding, we only focussed on the two dimensions that spanned the pitch circle. A wide range of linear and non-linear dimensionality reduction techniques – for example tSNE (*Maaten and Hinton, 2008*) – was tested leading to very similar results.

The weighted population decoder was computed by assigning each neuron its best pitch class (i.e. pitch class that evoked the highest response) and then evaluating the firing-rate weighted sum of all neurons' best pitch classes (see *Figure 4—figure supplement 1A* for a schematic). Since the stimulus space is circular, this weighted average was performed in the complex domain, where each neuron was represented by a unit vector in the complex plane, with an angle corresponding to the best pitch class. More precisely, this decoder is simply (omitting indices in the following)

$$\underline{PC}\left(S\right) = \sum_{i=1}^{Neurons} f_i\left(S\right)/max_{Stim}\left(f_i\right) \, * \, PC_{i,best}/P\left(PC_{i,best}\right), \, where \, PC_{i,best} \, \epsilon \, C$$

where $PC_{i,best}$ is the preferred pitch class of a cell and $C$ is the set of pitch classes, $f_i\left(S\right)$ the firing rate of the neuron $i$ for stimulus $S$. In the decoding, firing rate is normalized to the maximal firing rate for each cell, and the preferred pitch class for the empirical frequency of occurrence $P\left(PC_{i,best}\right)$, to compensate for uneven sampling of preferred pitch classes.

To assign a pitch class to the decoded stimuli of the test pair, we projected them onto the 'pitch-circle' formed by the decoded stimuli from the Bias sequences (*Figure 4A/B*). More precisely, we first estimated a coarse pitch circle with 24 steps at a resolution of 0.5 st, by averaging over bins of 10 neighboring pitch classes (partitioning the total of 240 Bias tones). Next, a more finely resolved trajectory through the set of Bias-tones at a resolution of 0.05 st was created by linear interpolation. Then, the pitch class of the test tone was set to the pitch class of the closest point on the trajectory.

For the present purpose the decoder was not cross-validated within the Bias sequence data, because its purpose was to provide a reference for the ambiguous pair stimuli, which were not part of the training set.

## Neural modeling

We used rate-based models of neural responses in the auditory cortex to investigate the link between the Bias-induced changes in response characteristic and the population decoding results. These are not trivially related, as different kinds of adaptation can lead to different – repulsive or attractive – effects (*Seriès et al., 2009*; *Jin et al., 2005*). Two types of models were investigated for this purpose:

(i) a *non-dynamic tuning model*, which serves to investigate generally the effect of different types of adaptation on the represented stimuli (see *Figure 5—figure supplement 1*).

## Non-dynamic neural models

In the non-dynamic model each neuron is represented as a *von Mises* distribution

$$M_i\left(S\,|\,\phi_i, \sigma_i\right) \;=\; exp\left(cos\left(2\pi/12\left(S - \phi_i\right)\right)/\left(2\pi/12\,\sigma_i\right)^2\right)/M_{total}$$

with two parameters, best pitch class $\phi_i$ and standard deviation $\sigma_i$, both measured in semitones, and $M_{total}$ normalizing the response to an area of 1. We simulate the response of a population of N=100 cortical neurons with $\phi_i$ equally spaced within [0,12] st. The models were run at the same sampling rate (20 Hz) as the data analysis for consistency.

The influence of the Bias is modeled assuming an idealized, continuous range of Biases, rather than individual tones. We consider three different models of adaptation: (a) local adaptation (*Figure 5—figure supplement 1A*) (b) global adaptation (*Figure 5—figure supplement 1B*), and (c) local adaptation with spreaded representation (*Figure 5—figure supplement 1C*):

a) *Local adaptation* refers to a multiplicative reduction of responses to individual stimuli, based on the local, recent stimulus history. The amount of local adaptation is taken as the prominence of this stimulus in the recent history, that is

$$A_i\left(\phi\,|\,S_{Bias}\right) \;=\; A_0 S_{Bias}\left(\phi\right)$$

where $S_{Bias}(\varphi)$ is defined as a function over [0,12] st taking values in [0,1]. $A_0$ is the maximal fraction of adaptation, set to 0.8 in Fig.S3. The cells adapted/biased response to a single Shepard tone is then given by

$$R_i\left(S\right) \;=\; \left(1 - A_i\left(S\,|\,S_{Bias}\right)\right) M_i\left(S\right)$$

In the more general case of a complex stimulus $S$, one would replace $M_i\left(S\right)$ with $M_i\left(S\right) * S$, that is the convolution of response and stimulus distribution.

This form of local adaptation resembles a highly stimulus specific version of adaptation. Hence, the responses are adapted only to previously presented stimuli, but no transfer to other stimuli occurs (see *Figure 5—figure supplement 1A*). This type of local adaptation leads to no adaptation, since neurons uniformly reduce their response to the test stimulus, which keeps the mean of the population response the same.

b) *Global adaptation* refers to a multiplicative reduction of the *entire* tuning curve, based on the recent response history, irrespective of which stimulus caused it. The amount of global adaptation is computed as the correlation between a cell's tuning curve and the stimulus history $S_{Bias}(PC)$, that is

$$A_i\left(S_{Bias}\right) \;=\; A_0\, S_{Bias} * R_i \;=\; \frac{1}{12}\int_0^{12} S_{Bias}\left(\phi\right) M_i\left(\phi\right)\,d\phi,$$

where * denotes convolution. By the normalization of both $S_{Bias}$ and $R_i$, $A_i$ will also be normalized within [0,$A_0$]. The cells biased tuning curve to a single Shepard tone is then given by

$$R_i\left(S\right) \;=\; \left(1 - A_i\left(S_{Bias}\right)\right) M_i\left(S\right)$$

Global adaptation in this sense captures summarized adaptation effects that occur 'globally' for the postsynaptic cell (e.g. a change in excitability which changes the slope of the IF-curve; see *Figure 5—figure supplement 1B*). Global adaptation shifts subsequent stimuli away from the adapting stimulus, since neurons close to the adaptor adapt more strongly (*Seriès et al., 2009*).

c) *Local adaptation with input spread* combines local adaptation with a distributed neural representation of pointlike stimuli (like a single tone or single Shepard tone), that is the stimulus is first represented on an intermediate level (e.g. the MGB) and then integrated on the cortical level, with adaptation occurring locally at the synapses connecting MGB-AI (see *Figure 5A* for an illustration of the architecture). Concretely, the cortical response is given as

$$R_{i,biased}\left(S\right) = \frac{1}{J}\sum_{j=1}^{J} M_i\left(j\right)\left(1 - A_i\left(j\right)\right) T_j\left(S\right)$$

where the intermediate representation $T_j\left(S\right)$ is given as

$$T_j\left(S\right) = S * M_j\left(S\right) = \frac{1}{12}\int_0^{12} M_j\left(\phi\right) S\left(\phi\right) d\phi$$

that is the convolution of the stimulus with the intermediate response properties $M_j\left(\phi\right)$ assumed to also be given by *von Mises* distributions as well. The adaptation $A_i\left(\phi\right)$ is equated with the midlevel activity induced by the bias

$$A_i\left(j|S_{Bias}\right) = A_0 T_j\left(S_{Bias}\right),$$

This form of local adaptation is 'less local' than the purely local adaptation described above. Hence, a presentation of a given stimulus will adapt the neural response not only to this stimulus, but – via the distributed representation – also for neighboring stimuli (see *Figure 5—figure supplement 1C*), which in the decoding leads to repulsive shifts, while reducing tuning curves locally. The resulting shape of the adaptation has been described as a shift in tuning curve combined with a global adaptation (*Jin et al., 2005*). We propose that the adaptation proposed here provides a simpler explanation for this observed shape of tuning curve change.

Note that the differences in decoding emerge only at the boundaries of the Bias region, depicted by the encoding-decoding matrices in *Figure 5—figure supplement 1* A3/B3/C3. If the distribution at the vertical line (at 0) has more weight above 0 on the abscissa, this corresponds to a repulsive shift.

In summary, purely local adaptation can account for the local changes in Shepard tunings of the real data, but fails to explain the repulsive decoding (*Figure 5—figure supplement 1A*). Global adaptation is consistent with the repulsive decoding results, but fails to explain the local tuning curve changes (*Figure 5—figure supplement 1B*). The combination of local adaptation and distributed input on the intermediate level (*Figure 5—figure supplement 1C*, *Figure 4*) is consistent with both the encoding and decoding findings.

This model is detailed in the Supplementary Methods and results are shown in *Figure 5—figure supplement 1*.

(ii) a *dynamic model*, which serves to use the insights of the non-dynamic model to account in more detail for the neural data. We used the identical stimulus sequences and analyses as for the real data. The structure of the dynamic model corresponded to non-dynamic model (c) (see above), that is a distributed stimulus representation before cortex and local adaptation in the thalamo-cortical synapses and (see *Figure 5A* for a schematic representation of the model). A sampling rate of 20 Hz was used for the simulations to speed up computations. Stimuli were represented as spectrograms – that is time-frequency representations – with 'frequency' being encoded as Shepard tones, that is they ranged over one octave and wrapped at the spectral boundaries.

In the mid-level (e.g. MGB) neural representation of the stimulus, each cell's response was modeled by a peak-normalized *von Mises* distribution for each time t of the filter, that is $M_i\left(\phi|\phi_i, \sigma_i\right) = R_{max}\, exp\left(cos\left(2\pi/12\left(\phi - \phi_i\right)\right) / \left(2\pi/12\,\sigma_i\right)^2\right) / M_{total}$, where $\phi$ denotes the stimulus, $\phi_i$ the mean of the distribution μ denotes the best pitch class and $\sigma$ the standard deviation, all in semitones. The maximal rate $R_{max}$ was arbitrarily set to 1, after normalizing the height to 1 by division via $M_{total}$. Hence, the responses on the mid-level $T_j\left(S\left(t\right)\right)$ of each neuron $j$ were modeled as a weighted average of the spectrogram at time $t$ with the neuron's tuning curve

$$T_j\left(S\left(t\right)\right) = \frac{1}{12}\sum_{\phi=1}^{12} M_j\left(\phi\right) S\left(t, \phi\right)$$

On the top-level, corresponding to auditory cortex, the activity of each neuron was modeled as a spectrotemporal filter on the activity of the mid-level representation with local synaptic depression at the synapses

$$R_i\left(S\left(t\right)\right) = \frac{1}{J}\sum_{j=1}^{J}\sum_{\tau=0}^{T-1} SSTRF_i\left(\tau,j\right)\left(1 - A_{i,j}\left(t\right)\right)T_j\left(S\left(t-\tau\right)\right)$$

where the $SSTRF_i\left(\tau,j\right)$ is the time-frequency filter for cortical neuron i, weighting the activity of the MGB neurons $j$ at times $\tau$=0...T before the current time. $SSTRF$ stands for Shepard Spectro-Temporal Receptive Field, which is equivalent to a classical STRF (**Depireux et al., 2001**), just for Shepard tones. The state of synaptic depression between cortical neuron $i$ and thalamic neuron $j$ is given by $A_{i,j}\left(t\right)$. The adaptation was determined by the activity locally present at each synapse and thus led to relatively local changes in the postsynaptic tuning curves. The dynamics of $A_{i,j}\left(t\right)$ are given by

$$A_{i,j}\left(t+1\right) = A_{i,j}\left(t\right)\left(1 - \boldsymbol{F_A}\sum_{\tau=0}^{T} SSTRF\left(\tau,j\right)/max\left(SSTRF\right)T_j\left(S\left(t-\tau\right)\right)\right)\left(1 - F_R\right)$$

where $\boldsymbol{F_A}$ is a constant weighting factor, which scales the amount of adaptation. In both cases the response computed via the $SSTRF$ is weighted with the adaptation coefficients $A_{PC/G}$, and each coefficient recovers by a fraction $\boldsymbol{F_R}$ in each step (leading to exponential recovery).

For the final simulations (**Figure 6**), the model was extended to contain a subset of directional cells, by extending the dependence of the $SSTRF$ by another 150ms (3 timesteps at the SR). A directional preference was implemented by adding a von Mises distribution (see above for definition) at the time range 150–250ms with a peak size of 0.25, roughly matching the observed peak-sizes in the $SSTRF$s of real directional cells. For downward preferring cells the center of the von Mises was placed relatively higher than the best semitone of the cell, and vice versa for upward preferring cells, in each case wrapping at the edges to account for the circularity of the Shepard tone response. The simulated population of 500 cells was split into one third non-directional cells, one third upward selective cells and one third downward selective cells.

## Tuning curve adaptation analysis

We estimated the *biased* Shepard tunings from the long stimulus sequences (see Acoustic Stimuli: *Biased Shepard Tuning*) by averaging the test stimuli for each location in the octave (see **Figure 4C**, different colors indicate different locations of the Bias sequence). To get an estimate of the unadapted tuning curve, we collected the initial 5 stimuli from each condition and thus constructed a corresponding tuning curve at a resolution of 1st. To evaluate the influence of the Bias, the local difference (**Figure 5D**) and fraction (**Figure 5E**) between the adapted and the unadapted tuning curve were analyzed. The same analysis was applied to model data generated from the identical stimuli using the same model as above (local adaptation, distributed input on the intermediate level, see **Figure 5— figure supplement 1C**).

## Directionality analysis

We investigated the effect of the Bias sequence on directionally selective cells. For this purpose, each cell's directional selectivity was estimated from the steps contained in the biasing sequences. $SSTRF$ were approximated by reverse correlating each neuron's response with the Bias sequences of Shepard tones (using normalized linear regression, three examples are shown in **Figure 6B**). The $SSTRF$s were discretized at 50ms, and include only the bins during the stimulus, not during the pause.

First, directional preference was assessed by computing the asymmetry in response strength in the second time bin $t_2$, centered on the maximal response in the first time bin, that is $DI = \sum_{\Delta<0}^{S} STRF\left(t_2, PC_{best} + \Delta\right) - \sum_{\Delta>0}^{S} STRF\left(t_2, PC_{best} + \Delta\right)$.

Positive values of $DI$ indicate up-ward selective cells and vice versa. The $SSTRF$ was first normalized to the maximal value to obtain comparable values between cells.

Second, a cell's spectral location relative to the test stimulus was determined by computing the distance between a cell's $SSTRF$ center-of-mass and the pitch class of the test tone. These first two steps, located a cell on the x- and y-axis of the following analysis (see **Figure 6C/D**, top).

Third, the difference in response for identical test stimuli with different preceding Bias locations (relative to each tone in the pair) was computed ('above' - 'below').

Finally, these differences were averaged for all cells with a given directionality and pitch-class relative to the target tone.

This analysis was also applied for the second test-stimulus, which means that each cell contributes to two locations, separated by the semi-octave distance between the two test-tones; however, the contribution was constituted by different (later) responses of the cell. This analysis was conducted both for the actual neural data, as well as for model data. These modeling results were obtained with the same model as above (local adaptation, distributed input on the intermediate level), although adaptation was set to 0 in one condition to demonstrate its role in generating the asymmetry of responses.

## Directional decoder

The decoder above first estimated the neurally represented pitch class of the stimulus and then evaluated the circular distance between the pitch classes for a given ambiguous pair to predict the percept. This approach implicitly assumes that the neural system organizes the neural representations correspondingly and can compute distances in this way. A more general and direct way of assessing whether a given stimulus is relatively higher or lower in pitch than a preceding stimulus may be to integrate the responses of neurons with regards to their directional preference (see previous section). We refer to this as the *directional hypothesis*. Quantitatively, this approach for decoding simply takes estimated direction selectivity $DI$ of each cell, and weighs it by its activity, and then sums across all cells. Analogously to decoding based on preferred pitch class, it thus assumes that a downstream decoder in the brain 'knows' about one or multiple characteristic properties of the cell (e.g. spectral and/or direction selectivity) and combines the activity of many cells in a weighted manner to arrive at a single estimate. We assume that this directionality is evaluated at the location of the current stimulus, that is the contribution of each cell is therefore weighted by the distance in preferred pitch class to the pitch class of the currently presented stimulus (see *Figure 6A* for the mathematical structure of the decoding).

## Tuning halfwidth

A neuron's tuning halfwidth with respect to Shepard tones was estimated using the range of Shepard tones that the firing rate was above $f_{50\%} = (f_{Max} - f_{Min})/2$. We used a conservative estimation method by determining $f_{Min}$ and then computing the range between the closest crossing of $f_{50\%}$ above and below $f_{Min}$. In this way, neurons with a small difference between $f_{Max}$ and $f_{Min}$ were assigned comparatively large tuning halfwidths, corresponding to their less salient tuning.

## Statistical analysis

Non-parametric tests were used throughout the study to avoid assumptions regarding distributional shape. Single group medians were assessed with the Wilcoxon signed rank test, two group median comparisons with the Mann-Whitney U-test, multiple groups with the Kruskal-Wallis (one-way) and Friedman test (two-way), with post-hoc testing performed using Bonferroni-correction of p-values. All tests are implemented in the Matlab Statistics Toolbox (The Mathworks, Natick).

## Acknowledgements

The authors would like to thank Barak Shechter, John Rinzel and Romain Brette for interesting discussions and comments on the manuscript. Funding information: European Research Council (Neume to SS); National Institutes of Health (to MH and SS: U01 AG058532). BE acknowledges funding from an NWO VIDI grant (016.VIDI.189.052) and a NWO ALW Open (ALWOP.146).

## Additional information

### Funding

| Funder | Grant reference number | Author |
| --- | --- | --- |
| Nederlandse Organisatie voor Wetenschappelijk Onderzoek | 016.VIDI.189.052 | Bernhard Englitz |

| Funder | Grant reference number | Author |
|---|---|---|
| National Institutes of Health | U01 AG058532 | Mounya Elhilali<br>Shihab Shamma |
| Deutsche Forschungsgemeinschaft | EN919/1-1 | Bernhard Englitz |
| European Research Council | Neume | Shihab Shamma |

The funders had no role in study design, data collection and interpretation, or the decision to submit the work for publication.

## Author contributions

Bernhard Englitz, Conceptualization, Data curation, Software, Funding acquisition, Investigation, Visualization, Methodology, Writing – original draft, Writing – review and editing; Sahar Akram, Data curation, Software, Investigation, Methodology, Writing – review and editing; Mounya Elhilali, Supervision, Writing – review and editing; Shihab Shamma, Conceptualization, Supervision, Funding acquisition, Project administration, Writing – review and editing

## Author ORCIDs

Bernhard Englitz ⓘ https://orcid.org/0000-0001-9106-0356
Mounya Elhilali ⓘ https://orcid.org/0000-0003-2597-738X

## Ethics

The human experiments were approved by the Institutional Review Board of the University of Maryland, College Park under the project number 1467378 ("The Adaptive Auditory Mind"). Listeners read and signed a consent form regarding data use and processing before data collection.

This study was performed in strict accordance with the recommendations in the Guide for the Care and Use of Laboratory Animals of the National Institutes of Health. All of the animals were handled according to approved institutional animal care and use committee (IACUC) protocols (Neurobiology of Auditory Cognition in Ferrets R-APR-23-24) of the University of Maryland, College Park.

Reviewer #1 (Public review): https://doi.org/10.7554/eLife.94296.3.sa1
Reviewer #2 (Public review): https://doi.org/10.7554/eLife.94296.3.sa2
Author response https://doi.org/10.7554/eLife.94296.3.sa3

# Additional files

## Supplementary files

• MDAR checklist

## Data availability

The data and code for reproducing the current analyses and figures have been deposited in collection di.dcn.DSC_626840_0009_350 inside the open access data repository of the Radboud University and can be accessed under the DOI: https://doi.org/10.34973/c0je-x552 .

The following dataset was generated:

| Author(s) | Year | Dataset title | Dataset URL | Database and Identifier |
|---|---|---|---|---|
| Englitz A, Elhilali S | 2024 | Decoding contextual influences on auditory perception from primary auditory cortex | https://doi.org/10.34973/c0je-x552 | Radboud Data Repository di.dcn.DSC_626840_0009_350, 10.34973/c0je-x552 |

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
