## [Editor Report · eLife Assessment]

This **important** study explores the neural basis for a well known auditory illusion, often utilized in movie soundtracks, in which a sequence of two complex tones can be perceived as either rising or falling in pitch depending on the context in which they are presented. **Convincing** single-neuron data and analyses are presented to show that correlates of these pitch-direction changes are found in the ferret primary auditory cortex. While these findings provide an interesting link between cortical activity and perception, the manuscript could be clearer on the wider implications of the failure of traditional decoding models to account for these results.

---

## [Referee Report · Reviewer #1 (Public review)]

Summary:

Previous work demonstrated a strong bias in the percept of an ambiguous Shepard tone as either ascending or descending in pitch, depending on the preceding contextual stimulus. The authors recorded human MEG and ferret A1 single-unit activity during presentation of stimuli identical to those used in the behavioral studies. They used multiple neural decoding methods to test if context-dependent neural responses to ambiguous stimulus replicated the behavioral results. Strikingly, a decoder trained to report stimulus pitch produced biases opposite to the perceptual reports. These biases could be explained robustly by a feed-forward adaptation model. Instead, a decoder that took into account direction selectivity of neurons in the population was able to replicate the change in perceptual bias.

Strengths:

This study explores an interesting and important link between neural activity and sensory percepts, and it demonstrates convincingly that traditional neural decoding models cannot explain percepts. Experimental design and data collection appear to have been executed carefully. Subsequent analysis and modeling appear rigorous. The conclusion that traditional decoding models cannot explain the contextual effects on percepts is quite strong.

Weaknesses:

Beyond the very convincing negative results, it is less clear exactly what the conclusion is or what readers should take away from this study. The presentation of the alternative, "direction aware" models is unclear, making it difficult to determine if they are presented as realistic possibilities or simply novel concepts. Does this study make predictions about how information from auditory cortex must be read out by downstream areas? There are several places where the thinking of the authors should be clarified, in particular, around how this idea of specialized readout of direction-selective neurons should be integrated with a broader understanding of auditory cortex.

---

## [Referee Report · Reviewer #2 (Public review)]

Summary:

This is an elegant study investigating possible mechanisms underlying the hysteresis effect in the perception of perceptually ambiguous Shepard tones. The authors make a fairly convincing case that the adaptation of pitch direction sensitive cells in auditory cortex is likely responsible for this phenomenon.

Strengths:

The manuscript is overall well written. My only slight criticism is that, in places, particularly for non-expert readers, it might be helpful to work a little bit more methods detail into the results section, so readers don't have to work quite so hard jumping from results to methods and back.

The methods seem sound and the conclusions warranted and carefully stated. Overall I would rate the quality of this study as very high, and I do not have any major issues to raise.

Weaknesses:

I think this study is about as good as it can be with the current state of the art. Generally speaking, one has to bear in mind that this is an observational, rather than an interventional study, and therefore only able to identify plausible candidate mechanisms rather than making definitive identifications. However, the study nevertheless represents a significant advance over the current state of knowledge, and about as good as it can be with the techniques that are currently widely available.

---

## [Author Response]

The following is the authors’ response to the original reviews.

**Reviewer #1 (Public Review):**
Summary:Previous work demonstrated a strong bias in the percept of an ambiguous Shepard tone as either ascending or descending in pitch, depending on the preceding contextual stimulus. The authors recorded human MEG and ferret A1 single-unit activity during presentation of stimuli identical to those used in the behavioral studies. They used multiple neural decoding methods to test if context-dependent neural responses to ambiguous stimulus replicated the behavioral results. Strikingly, a decoder trained to report stimulus pitch produced biases opposite to the perceptual reports. These biases could be explained robustly by a feed-forward adaptation model. Instead, a decoder that took into account direction selectivity of neurons in the population was able to replicate the change in perceptual bias.Strengths:This study explores an interesting and important link between neural activity and sensory percepts, and it demonstrates convincingly that traditional neural decoding models cannot explain percepts. Experimental design and data collection appear to have been executed carefully. Subsequent analysis and modeling appear rigorous. The conclusion that traditional decoding models cannot explain the contextual effects on percepts is quite strong.Weaknesses:Beyond the very convincing negative results, it is less clear exactly what the conclusion is or what readers should take away from this study. The presentation of the alternative, "direction aware" models is unclear, making it difficult to determine if they are presented as realistic possibilities or simply novel concepts. Does this study make predictions about how information from auditory cortex must be read out by downstream areas? There are several places where the thinking of the authors should be clarified, in particular, around how this idea of specialized readout of direction-selective neurons should be integrated with a broader understanding of auditory cortex.

While we have not used the term "direction aware", we think the reviewer refers generally to the capability of our model to use a cell's direction selectivity in the decoding. In accordance with the reviewer's interpretation, we did indeed mean that the decoder assumes that a neuron does not only have a preferred frequency, but also a preferred direction of change in frequency (ascending/descending), which is what we use to demonstrate that the decoding in this way aligns with the human percept. We have adapted the text in several places to clarify this, in particular expanding the description in the Methods substantially.

**Reviewer #2 (Public Review):**
The authors aim to better understand the neural responses to Shepard tones in auditory cortex. This is an interesting question as Shepard tones can evoke an ambiguous pitch that is manipulated by a proceeding adapting stimulus, therefore it nicely disentangles pitch perception from simple stimulus acoustics.The authors use a combination of computational modelling, ferret A1 recordings of single neurons, and human EEG measurements.Their results provide new insights into neural correlates of these stimuli. However, the manuscript submitted is poorly organized, to the point where it is near impossible to review. We have provided Major Concerns below. We will only be able to understand and critique the manuscript fully after these issues have been addressed to improve the readability of the manuscript. Therefore, we have not yet reviewed the Discussion section.Major concernsOrganization/presentationThe manuscript is disorganized and therefore difficult to follow. The biggest issue is that in many figures, the figure subpanels often do not correspond to the legend, the main body, or both. Subpanels described in the text are missing in several cases.

We have gone linearly through the text and checked that all figure subpanels are referred to in the text and the legend. As far as we can tell, this was already the case for all panels, with the exception of two subpanels of Fig. 5.

Many figure axes are unlabelled.

We have carefully checked the axes of all panels and all but two (Fig. 5D) were labeled. As is customary, certain panels inherit the axis label from a neighboring panel, if the label is the same, e.g. subpanels in Fig. 6F or Fig. 5E, which helps to declutter the figure. We hope that with this clarification, the reviewer can understand the labels of each panel.

There is an inconsistent style of in-text citation between figures and the main text. The manuscript contains typos and grammatical errors. My suggestions for edits below therefore should not be taken as an exhaustive list. I ask the authors to consider the following only a "first pass" review, and I will hopefully be able to think more deeply about the science in the second round of revisions after the manuscript is better organized.

While we are puzzled by the severity of issues that R2 indicates (see above, and R3 qualifies it as "well written", and R1 does not comment on the writing negatively), we have carefully gone through all specific issues mentioned by R2 and the other reviewers. We hope that the revised version of the paper with all corrections and clarifications made will resolve any remaining issues.

Frequency and pitchThe terms "frequency" and "pitch" seem to be used interchangeably at times, which can lead to major misconceptions in a manuscript on Shepard tones. It is possible that the authors confuse these concepts themselves at times (e.g. Fig 5), although this would be surprising given their expertise in this field. Please check through every use of "frequency" and "pitch" in this manuscript and make sure you are using the right term in the right place. In many places, "frequency" should actually be "fundamental frequency" to avoid misunderstanding.

Thanks for pointing this out. We have checked every occurrence and modified where necessary.

Insufficient detail or lack of clarity in descriptionsThere seems to be insufficient information provided to evaluate parts of these analysis, most critically the final pitch-direction decoder (Fig 6), which is a major finding. Please clarify.

Thanks for pointing this out. We have extended the description of the pitch-direction decoder and highlighted its role for interpreting the results.

**Reviewer #3 (Public Review):**
Summary:This is an elegant study investigating possible mechanisms underlying the hysteresis effect in the perception of perceptually ambiguous Shepard tones. The authors make a fairly convincing case that the adaptation of pitch direction sensitive cells in auditory cortex is likely responsible for this phenomenon.Strengths:The manuscript is overall well written. My only slight criticism is that, in places, particularly for non-expert readers, it might be helpful to work a little bit more methods detail into the results section, so readers don't have to work quite so hard jumping from results to methods and back.

Following this excellent suggestion, we have added more brief method sketches to the Results section, hopefully addressing this concern.

The methods seem sound and the conclusions warranted and carefully stated. Overall I would rate the quality of this study as very high, and I do not have any major issues to raise.

Thanks for your encouraging evaluation of the work.

Weaknesses:I think this study is about as good as it can be with the current state of the art. Generally speaking, one has to bear in mind that this is an observational, rather than an interventional study, and therefore only able to identify plausible candidate mechanisms rather than making definitive identifications. However, the study nevertheless represents a significant advance over the current state of knowledge, and about as good as it can be with the techniques that are currently widely available.

Thanks for your encouraging evaluation of our work. The suggestion of an interventional study has also been on our minds, however, this appears rather difficult, as it would require a specific subset of cells to be inhibited. The most suitable approach would likely be 2p imaging with holographic inhibition of a subset of cells (using ArchT for example), that has a preference for one direction of pitch change, which should then bias the percept/behavior in the opposite direction.

**Reviewer #1 (Recommendations For The Authors):**
MAJOR CONCERNS(1) What is the timescale used to compute direction selectivity in neural tuning? How does it compare to the timing of the Shepard tones? The basic idea of up versus down pitch is clear, the intuition for the role of direction tuning and its relation to stimulus dynamics could be laid out more clearly. Are the authors proposing that there are two "special" populations of A1 neurons that are treated differently to produce the biased percept? Or is there something specific about the dynamics of the Shepard stimuli and how direction selective neurons respond to them specifically? It would help if the authors could clarify if this result links to broader concepts of dynamic pitch coding in general or if the example reported here is specific (or idiosyncratic) to Shepard tones.

We propose that the findings here are not specific to Shepard tones. To the contrary, only basic properties of auditory cortex neurons, i.e. frequency preference, frequency-direction (i.e. ascending or descending) preference, and local adaptation in the tuning curve, suffice. Each of these properties have been demonstrated many times before and we only verified this in the lead-up to the results in Fig. 6. While the same effects should be observable with pure tones, the lack of ambiguity in the perception of direction of a frequency step for pure tone pairs, would make them less noticeable here. Regarding the time-scale of the directional selectivity, we relied on the sequencing of tones in our paradigm, i.e. 150 ms spacing. The SSTRFs were discretized at 50 ms, and include only the bins during the stimulus, not during the pause. The directional tuning, i.e. differences in the SSTRF above and below the preferred pitchclass for stimuli before the last stimulus, typically extended only one stimulus back in time. We have clarified this in more detail now, in particular in the added Methods section on the directional decoder.

(2) (p. 9) "weighted by each cell's directionality index ... (see Methods for details)" The direction-selective decoder is interesting and appears critical to the study. However, the details of its implementation are difficult to locate. Maybe Fig. 6A contains the key concepts? It would help greatly if the authors could describe it in parallel with the other decoders in the Methods.

We have expanded the description of the decoder in the Methods as the reviewer suggests.

LESSER CONCERNSp. 1. (L 24) "distances between the pitch representations...." It's not obvious what "distances" means without reading the main paper. Can some other term or extra context be provided?

We have added a brief description here.

p. 2. (L 26) "Shepard tones" Can the authors provide a citation when they first introduce this class of stimuli?

Citation has been added.

p. 3 (L 4) "direction selective cells" Please define or provide context for what has a direction. Selective to pitch changes in time?

Yes, selective to pitch changes in time is what is meant. We have further clarified this in the text.

p. 4 (L 9-19). This paragraph seems like it belongs in the Introduction?

Given the concerns raised by R2 about the organization of the manuscript we prefer to keep this 'road-map' in the manuscript, as a guidance for the reader.

p. 4 (L 32) "majority of cells" One might imagine that the overlap of the bias band and the frequency tuning curve of individual neurons might vary substantially. Was there some criterion about the degree of overlap for including single units in the analysis? Does overlap matter?

We are not certain which analysis the reviewer is referring to. Generally, cells were not excluded based on their overlap between a particular Bias band and their (Shepard) tuning curve. There are several reasons for this: The bias was located in 4 different, overlapping Shepard tone regions, and all sounds were Shepard tones. Therefore, all cells overlapped with their (Shepard) tuning curve with one or multiple of the Biases. For decoding analysis, all cells were included as both a response and lack of a response is contributing to the decoding. If the reviewer is referring only to the analysis of whether a cell adapts, then the same argument applies as above, i.e. this was an average over all Bias sequences, and therefore every responding cell was driven to respond by the Bias, and therefore it was possible to also assess whether it adapted its response for different positions inside the Bias. We acknowledge that the limited randomness of the Bias sequences in combination with the specific tuning of the cells could in a few cases create response patterns over time that are not indicative of the actual behavior for repeated stimulation, however, since the results are rather clear with 91% of cells adapting, we do not think this would significantly change the conclusions.

p. 5 (L 17) "desynchronization ... behaving conditions" The logic here is not clear. Is less desynchronization expected during behavior? Typically, increased attention is associated with greater desynchronization.

Yes, we reformulated the sentence to: While this difference could be partly explained by desynchronization which is typically associated with active behavior or attention [30], general response adaptation to repeated stimuli is also typical in behaving humans [31].

p. 7 (L 5) "separation" is this a separation in time?

Yes, added.

p. 7 (L 33) "local adaptation" The idea of feedforward adaptation biasing encoding has been proposed before, and it might be worth citing previous work. This includes work from Nelken specifically related to SSA. Also, this model seems similar to the one described in Lopez Espejo et al (PLoS CB 2019).

Thanks for pointing this out. We think, however, that neither of these publications suggested this very narrow way of biasing, which we consider biologically implausible. We have therefore not added either of these citations.

p. 11 (L. 17) The cartoon in Fig. 6G may provide some intuition, but it is quite difficult to interpret. Is there a way to indicate which neuron "votes" for which percept?

This is an excellent idea, and we have added now the purported perceptual relation of each cell in the diagram.

p. 12 (L. 8). "classically assumed" This statement could benefit from a citation. Or maybe "classically" is not the right word?

We have changed 'classically' to 'typically', and now cite classical works from Deutsch and Repp. We think this description makes sense, as the whole concept of bistable percepts has been interpreted as being equidistant (in added or subtracted semitone steps) from the first tone, see e.g. Repp 1997, Fig.2.

p. 12 (L. 12) "...previous studies" of Shepard tone percepts? Of physiology?

We have modified it to 'Relation to previous studies of Shepard tone percepts and their underlying physiology", since this section deals with both.

p. 12 (L. 25) "compatible with cellular mechanisms..." This paragraph seems key to the study and to Major Concern 1, above. What are the dynamics of the task stimuli? How do they compare with the dynamics of neural FM tuning and previously reported studies of bias? And can the authors be more explicit in their interpretation - should direction selective neurons respond preferentially to the Shepard tone stimuli themselves? And/or is there a conceptual framework where the same neurons inform downstream percepts of both FM sweeps and both normal (unbiased) and biased Shepard tones?

The reviewer raises a number of different questions, which we address below:

- Dynamics of the task stimuli in relation to previously reported cellular biasing: The timescales tested in the studies mentioned are similar to what we used in our bias, e.g. Ye et al 2010 used FM sweeps that lasted for up to 200ms, which is quite comparable to our SOA of 150ms.

- Preferred responses to Shepard tones: no, we do not think that there should be preferred responses to Shepard tones, but rather that responses to Shepard tones can be thought of as the combined responses to the constituent tones.

- Conceptual framework where the same neurons inform about FM sweeps and both normal (unbiased) and biased Shepard tones: Our perspective on this question is as follows: To our knowledge, the classical approach to population decoding in the auditory system, i.e. weighted based on preferred frequency, has not been directly demonstrated to be read out inside the brain, and certainly not demonstrated to be read out in only this way in all areas of the brain that receive input from the auditory cortex. Rather it has achieved its credibility by being linked directly with animal performance or match with the presented stimuli. However, these approaches were usually geared towards a representation that can be estimated based on constituent frequencies. Additional response properties of neurons, such as directional selectivity have been documented and analyzed before, however, not been used for explaining the percept. We agree that our use of this cellular response preference in the decoding implicitly assumes that the brain could utilize this as well, however, this seems just as likely or unlikely as the use of the preferred frequency of a neuron. Therefore we do not think that this decoding is any more speculative than the classical decoding. In both cases, subsequent neurons would have to implicitly 'know' the preference of the input neuron, and weigh its input correspondingly.

We have added all the above considerations to the discussion in an abbreviated form.

p. 15 (L. 15). Is there a citation for the drive system?

There is no publication, but an old repository, where the files are available, which we cite now: https://code.google.com/archive/p/edds-array-drive/

p. 16 (L. 24) "position in an octave" It is implied but not explicitly stated that the Shepard tones don't contain the fundamental frequency. Can the authors clarify the relationship between the neural tuning band and the bands of the stimulus. Did a single stimulus band typically fall in a neuron's frequency tuning curve? If not 1, how many?

Yes, it is correct that the concept of fundamental frequency does not cleanly apply to Shepard tones, because it is composed of octave spaced pure tones, but the lowest tone is placed outside the hearing range of the animal and amplitude envelope (across frequencies). Therefore one or more constituent tones of the Shepard tone can fall into the tuning curve of a neuron and contribute to driving the neuron (or inhibiting it, if they fall within an inhibitory region of the tuning curve). The number of constituent tones that fall within the tuning curve depends on the tuning width of the neurons. The distribution of tuning widths to Shepard tones is shown in Fig. S1E, which indicated that a lot of neurons had rather narrow tuning (close to the center), but many were also tuned widely, indicated that they would be stimulated by multiple constituent tones of the Shepard tone. As the tuning bandwidth (Q30: 30dB above threshold) of most cortical neurons in the ferret auditory cortex (see e.g. Bizley et al. Cerebral Cortex, 2005, Fig.12) is below 1, this means that typically not more than 1 tone fell into the tuning curve of a neuron. However, we also observed multimodal tuning-curves w.r.t. to Shepard tones, which suggests that some neurons were stimulated by more than 2 or more constituent tones again consistent with the existence of more broadly tuned neurons (see same citation). We have added this information partly to the manuscript in the caption of Fig. S1E.

p. 17 (L. 32). "Fig 4" Correct figure ref? This figure appears to be a schematic rather than one displaying data.

Thanks for pointing this out, changed to Fig. 5.

p. 18 (L. 25). "assign a pitchclass" Can the authors refer to a figure illustrating this process?

Added.

p. 19 (L. 17). Is mu the correct symbol?

Thanks. We changed it to phi_i, as in the formula above.

p. 19 (L 19). "convolution" in time? Frequency?

Thanks for pointing this out, the term convolution was incorrect in this context. We have replaced it by "weighted average" and also adapted and simplified the formula.

p. 19 (L 25) "SSTRF" this term is introduced before it is defined. Also it appears that "SSTRF" and "STRF" are sometimes interchanged.

Apologies, we have added the definition, and also checked its usage in each location.

p. 23 (Fig 2) There is a mismatch between panel labels in the figure and in the legend. Bottom right panel (B3), what does time refer to here?

Thanks for pointing these out, both fixed.

p. 24 (L 23) "shifts them away" away from what?

We have expanded the sentence to: "After the bias, the decoded pitchclass is shifted from their actual pitchclass away from the biased pitchclass range ... "

p. 25 (L 7) "individual properties" properties of individual subjects?

Thanks for pointing this out, the corresponding sentence has been clarified and citations added.

p. 26 (L 20) What is plotted in panel D? The average for all cells? What is n?

Yes, this is an average over cells, the number of cells has now been added to each panel.

p. 28 (L 3) How to apply the terms "right" "right" "middle" to the panel is not clear. Generally, this figure is quite dense and difficult to interpret.

We have changed the caption of Panel A and replaced the location terms with the symbols, which helps to directly relate them to the figure. We have considered different approaches of adding or removing content from the figure to help make it less dense, but that all did not seem to help. For lack of better options we have left it in its current form.

MINOR/TYPOSp. 3 (L 1) "Stimulus Specific Adaptation" Capitalization seems unnecessary

Changed.

p. 4 (L 14) "Siple"

Corrected.

p. 9 (L 10) "an quantitatively"

Corrected

p. 9 (L 20) "directional ... direction ... directly ... directional" This is a bit confusing as directseems to mean several different things in its different usages.

We have gone through these sentences, and we think the terms are now more clearly used, especially since the term 'direction' occurs in several different forms, as it relates to different aspects (cells/percept/hypothesis). Unfortunately, some repetition is necessary to maintain clarity.

**Reviewer #2 (Recommendations For The Authors):**
Detailed critiqueStimuliIt would be very useful if the authors could provide demos of their stimuli on a website. Many readers will not be familiar with Shepard tones and the perceptual result of the acoustical descriptions are not intuitive. I ended up coding the stimuli myself to get some intuition for them.

We have created some sample tones and sequences and uploaded them with the revision as supplementary documents.

AbstractP1 L27 'pitch and...selective cells' - The authors haven't provided sufficient controls to demonstrate that these are "pitch cells" or "selective" to pitch direction. They have only shown that they are sensitive to these properties in their stimuli. Controls would need to be included to ensure that the cells aren't simply responding to one frequency component in the complex sound, for example. This is not really critical to the overall findings, but the claim about pitch "selectivity" is not accurate.

Fair point. We have removed the word 'selective' in both occurrences.

IntroductionP2 L14-17: I do not follow the phonetic example provided. The authors state that the second syllable of /alga/ and /arda/ are physically identical, but how is this possible that ga = da? The acoustics are clearly different. More explanation is needed, or a correction.

Apologies for the slightly misleading description, it has now been corrected to be in line with the original reference.

P2,L26-27: Should the two uses of "frequency" be "F0" and "pitch" here? The tones are not separated in frequency by half and octave, but "separated in [F0]" by half an octave, correct? Their frequency ranges are largely overlapping. And the second 'frequency', which refers to the percept, should presumably be "pitch".

Indeed. This is now corrected.

P3 L2-6: Unclear at this point in the manuscript what is the difference between the 3 percepts mentioned: perceived pitch-change direction, Shepard tone pitches, and "their respective differences". (It becomes clear later, but clarification is needed here).

We have tried a few reformulations, however, it tends to overload the introduction with details. We believe it is preferable to present the gist of the results here, and present the complete details later in the MS.

P3 L6-7 What does it mean that the MEG and single unit results "align in direction and dynamics"? These are very different signals, so clarification is needed.

We have phrased the corresponding sentence more clearly.

ResultsThroughout: Choose one of 'pitch class', 'pitchclass', or 'pitch-class' and use it consistently.

Done.

P4L12 - would be helpful at this point to define 'repulsive effect'

We have added another sentence to clarify this term.

P4, L14 "simple"

Done

P4, L12 - not clear here what "repulsive influence" means

See above.

P4, L17 - alternative to which explanation? Please clarify. In general, this paragraph is difficult to interpret because we do not yet have the details needed to understand the terms used and the results described. In my opinion, it would be better to omit this summary of the results at the very beginning, and instead reveal the findings as they come, when they can be fully explained to the Reader.

We agree, but we also believe that a rather general description here is useful for providing a roadmap to the results. However, we have added a half-sentence to clarify what is meant by alternative.

P4 L30 - text says that cells adapt in their onset, sustained and offset responses, but only data for onset responses are shown (I think - clarification needed for fig 2A2). Supp figure shows only 1 example cell of sustained and offset, and in fact there is no effect of adaptation in the sustained response shown there.

Regarding the effect of adaptation and whether it can be discerned from the supplementary figure: the shown responses are for 10 repetitions of one particular Bias sequence. Since the response of the cell will depend on its tuning and the specific sequence of the Shepard tones in this Bias, it is not possible to assess adaptation for a given cell. We assess the level of adaptation, by averaging all biases (similar to what is shown in Fig. 2A2) per cell, and then fit an exponential to it, separately by response type. The step direction of the exponential, relative to the spontaneous rate is then used to assess the kind of adaptation. The vast majority of cells show adaptation. We have added this information to the Methods of the manuscript.

P4, L32 - please state the statistical test and criterion (alpha) used to determine that 91% of cells decreased their responses throughout the Bias sequence. Was this specifically for onset responses?

Thanks for pointing this out, test and p-value added. Adaptation was observed for onset, sustained and offset responses, in all cases with the vast majority showing an adapting behavior, although the onset responses were adapting the most.

P4 L36 - "response strength is reduced locally". What does "locally" mean here? Nearby frequencies?

We have added a sentence here to clarify this question.

Figure 1 - this appears to be the wrong version of the figure, as it doesn't match the caption or results text. It's not possible to assess this figure until these things are fixed. Figure 1A schematic of definition of f(diff) does not correspond to legend definition.

As far as we can tell, it is all correct, only the resolution of the figure appears to be rather low. This has been improved now.

Fig 2 A2 - is this also onset responses only?

Yes, added to the caption.

Fig 2 A3 - add y-axis label. The authors are comparing a very wide octave band (5.5 octaves) to a much narrower band (0.5 octaves). Could this matter? Is there something special about the cut-off of 2.5 octaves in the 2 bands, or was this an arbitrary choice?

Interesting question.... essentially our stimulus design left us only with this choice, i.e. comparing the internal region of the bias with the boundary region of the bias, i.e. the test tones. The internal region just corresponds to the bias, which is 5 st wide, and therefore the range is here given as 2.5 st relative to its center, while the test tones are at the boundary, as they are 3 st from the center. The axis for the bias was mislabelled, and has now been corrected. The y-axis label is matched with the panel to the left, but has now been added to avoid any confusion.

Fig 2A4 - does not refer to ferret single unit data, as stated in the text (p5L8). Nor does supp Fig2, as stated. Also, the figure caption does not match the figure.

Apologies, this was an error in the code that led to this mislabelling. We have corrected the labels, which also added back the recovery from the Bias sequence in the new Panel A4.

P5 l9 - Figure 3 is not understandable at this point in the text, and should not be referred to here. There is a lot going on in Fig 3, and it isn't clear what you are referring to.

Removed.

P5 L12 - by Fig 2 B1, I assume you mean A4? Also, F2B1 shows only 1 subject, not 2.

Yes, mislabeled by mistake, and corrected now.

Fig2B2 -What is the y-axis?

Same as in the panel to its left, added for clarity.

Stimuli: why are tones presented at a faster rate to ferrets than to humans?

The main reason is that the response analysis in MEG requires more spacing in time than the neuronal analysis in the ferret brain.

P5 L6 - there is no Fig 5 D2? I don't think it is a good idea to get the reader to skip so far ahead in the figures at this stage anyway, even if such a figure existed. It is confusing to jump around the manuscript

Changed to 'see below'

P5 L8 - There is no Figure 2A4, so I don't know whether this time constant is accurate.

This was in reference to a panel that had been removed before, but we have added it back now.

P5 L16: "in humans appears to be more substantial (40%) than for the average single units under awake conditions". One cannot directly compare magnitude of effects in MEG and single unit signals in this way and assume it is due to behavioural state. You are comparing different measures of neural activity, averaged over vastly different numbers of numbers, and recorded from different species listening to different stimuli (presentation rates).

Yes, that's why the next sentence is: "However, comparisons between the level of adaptation in MEG and single neuron firing rates may be misleading, due to the differences in the signal measured and subsequent processing.", and all statements in the preceding sentences are phrased as 'appears' and 'may'. We think we have formulated this comparison with an appropriate level of uncertainty. Further, the main message here is that adaptation is taking place in both active and passive conditions.

P5 L25 -I do not see any evidence regarding tuning widths in Fig s2, as stated in the text.

Corrected to Fig. S1.

P5 l26 - Do not skip ahead to Fig 5 here. We aren't ready to process that yet.

OK, reference removed.

P5 l27 - Do you mean because it could be tuning to pitch chroma, not height?

Yes, that is a possible interpretation, although it could also arise from a combination of excitatory and inhibitory contributions across multiple octaves.

P5 l33 - remove speculation about active vs passive for reasons given above.

Removed.

P6L2-6 'In the present...5 semitone step' - This is an incorrect interpretation of the minimal distance hypothesis in the context of the Shepard tone ambiguity. The percept is ambiguous because the 'true' F0 of the Shepard tones are imperceptibly low. Each constituent frequency of a single tone can therefore be perceived either as a harmonic of some lower fundamental frequency or as an independent tone. The dominant pitch of the second tone in the tritone pair may therefore be biased to be perceived at a lower constituent frequency (when the bias sequence is low) or at a higher constituent frequency (when the bias sequence is high). The text states that the minimal distance hypothesis would predict that an up-bias would make a tritone into a perfect fourth (5 semitones). This is incorrect. The MDH would predict that an up-bias would reduce the distance between the 1st tone in the ambiguous pair and the upper constituent frequency of the 2nd tone in the pair, hence making the upper constituent frequency the dominant pitch percept of the 2nd tone, causing an ascending percept.

The reviewer here refers to a “minimal distance hypothesis”, which without a literature reference,is hard for us to fully interpret. However, some responses are given below:

- "The percept is ambiguous because the 'true' F0 of the Shepard tones are imperceptibly low." This statement appears to be based on some misconception: due to the octave spacing (rather than multiple/harmonics of a lowest frequency), the Shepard tones cannot be interpreted as usual harmonic tones would be. It is correct that the lowest tone in a Shepard tone is not audible, due to the envelope and the fact that it could in principle be arbitrarily small... hence, speaking about an F0 is really not well-defined in the case of a Shepard tone. The closest one could get to it would be to refer to the Shepard tone that is both in the audible range and in the non-zero amplitude envelope. But again, since the envelope is fading out the highest and lowest constituent tones, it is not as easy to refer to the lowest one as F0 as it might be much quieter than the next higher constituent.

- "The dominant pitch of the second tone in the tritone pair may therefore be biased to be perceived at a lower constituent frequency (when the bias sequence is low) or at a higher constituent frequency (when the bias sequence is high)." This may relate to some known psychophysics, but we are unable to interpret it with certainty.

- "The text states that the minimal distance hypothesis would predict that an up-bias would make a tritone into a perfect fourth (5 semitones). This is incorrect." We are unsure how the reviewer reaches this conclusion.

- "The MDH would predict that an up-bias would reduce the distance between the 1st tone in the ambiguous pair and the upper constituent frequency of the 2nd tone in the pair, hence making the upper constituent frequency the dominant pitch percept of the 2nd tone, causing an ascending percept." Again, in the absence of a reference to the MDH, we are unsure of the implied rationale. We agree that this is a possible interpretation of distance, however, we believe that our interpretation of distance (i.e. distances between constituent tones) is also a possible interpretation.

Fig 4: Given that it comes before Figure 3 in the results text, these should be switched in order in the paper.

Switched.

PCA decoder: The methods (p18) state that the PCA uses the first 3 dimensions, and that pitch classes are calculated from the closest 4 stimuli. The results (P6), however, state that the first 2 principal components are used, and classes are computed from the average of 10 adjacent points. Which is correct, or am I missing something?

Thanks for pointing this out, we have made this more concrete in the Methods to: "The data were projected to the first three dimensions, which represented the pitch class as well as the position in the sequence of stimuli (see Fig. 43A for a schematic). As the position in the Bias sequence was not relevant for the subsequent pitch class decoding, we only focussed on the two dimensions that spanned the pitch circle." Regarding the number of stimuli that were averaged: this might be a slight misunderstanding: Each Shepard tone was decoded/projected without averaging. However, to then assign an estimated pitch class, we first had to establish an axis (here going around the circle), where each position along the axis was associated with a pitch class. This was done by stepping in 0.5 semitone steps, and finding the location in decoded space that corresponded to the median of the Shepard tones within +/- 0.25st. To increase the resolution, this circular 'axis' of 24 points was then linearly interpolated to a resolution of 0.05st. We have updated the text in the Methods accordingly. The mentioning of 10 points for averaging in the Results was correct, as there were 240 tones in all bias stimuli, and 24 bins in the pitch circle. The mentioning of an average over 4 tones in the Methods was a typo.

Fig 3A: axes of pink plane should be PC not PCA

Done.

Fig 3B: the circularity in the distribution of these points is indeed interesting! But what do the authors make of the gap in the circle between semitones 6-7? Is this showing an inherent bias in the way the ambiguous tone is represented?

While we cannot be certain, we think that this represents an inhomogeneous sampling from the overall set of neural tuning preferences, and that if we had recorded more/all neurons, the circle would be complete and uniformly sampled (which it already nearly is, see Fig.4C, which used to be Fig. 3C).

Fig 3B (lesser note): It'd be preferable to replace the tint (bright vs. dark) differentiation of the triangles to be filled vs. unfilled because such a subtle change in tint is not easily differentiable from a change in hue (indicating a different variable in this plot) with this particular colour palette

We have experimented with this suggestion, and it didn't seem to improve the clarity. However, we have changed the outline of the test-pair triangles to white, which now visually separates them better.

P6 l32 - Please indicate if cross-validation was used in this decoder, and if so, what sort. Ideally, the authors would test on a held-out data set, or at least take a leave-one-out approach. Otherwise, the classifier may be overfit to the data, and overfitting would explain the exceptional performance (r=.995) of the classifier.

Cross-validation was not used, as the purpose of the decoder is here to create a standard against which to compare the biased responses in the ambiguous pair, which were not used for training of the decoder. We agree that if we instead used a cross-validated decoder (which would only apply to the local average to establish the pitch class circle) the correlation would be somewhat lower, however, this is less relevant for the main question, i.e. the influence of the Bias sequence on the neural representation of the ambiguous pair. We have added this information to the corresponding section.

Fig 3D: I understood that these pitch classifications shown by the triangles were carried out on the final ambiguous pair of stimuli. I thought these were always presented at the edges of the range of other stimuli, so I do not follow how they have so many different pitchclass values on the x-axis here.

There were 4 Biases, centered at 0,3,6 or 9 semitones, and covering [-2.5,2.5]st relative to this center. Therefore the edges of the bias ranges (3st away from their centers) happen to be the same as the centers, e.g. for the Bias centered at 3, the ambiguous pair would be a 0-6 or 6-0 step. Therefore there are 4 locations for the ambiguous tones on the x-axis of Fig. 4D (previously 3D).

Figure 4: This demonstration of the ambiguity of Shepard pairs may be misleading. The actual musical interval is never ambiguous, as this figure suggests. Only the ascending vs descending percept is ambiguous. Therefore the predictions of the ferret A1 decoding (Fig 3D) and the model in Fig 5 are inconsistent with perception in two ways. One (which the authors mention) is the direction of the bias shift (up vs down). Another (not mentioned here) is that one never experiences a shift in the shepard tone at a fraction of a semitone - the musical note stays the same, and changes only in pitch height, not pitch chroma.

We are unsure of the reviewer’s direction with this question. In particular the second point is not clear to us: "...one (who?) never (in this experiment? in real life?) experiences a bias shift in the Shepard tone at a fraction of a semitone" (why is this relevant in the current experiment?). Pitch chrome would actually be a possible replacement for pitch class, but somehow, the previous Shepard tone literature has referred to it as pitch class.

P7 l12 - omit one 'consequently'

Changed to 'Therefore'.

P7 l24 - I encourage the authors to not use "local" and "global" without making it clear what space they refer to. One tends to automatically think of frequency space in the auditory system, but I think here they mean f0 space? What is a "cell close to the location of the bias"? Cells reside in the brain. The bias is in f0 space. The use of "local" and "global" throughout the manuscript is too vague.

Agreed, the reference here was actually to the cell's preferred pitch class, not its physical location (which one might arguably be able to disambiguate, given the context). We have changed the wording, and also checked the use of global/local throughout the manuscript. The main use of 'global/local' is now in reference to the range of adaptation, and is properly introduced on first mention.

P7 L26 -there is no Fig 5D1. Do you mean the left panel of 5D?

Thanks. Changed.

FigS3 is referred to a lot on p7-8. Should this be moved to the main text?

The main reason why we kept it in the supplement is that it is based on a more static model, which is intended to illustrate the consequences of different encoding schemes. In order to not confuse the reader about these two models, we prefer to keep it in the supplement, which - for an online journal - makes little difference since the reader can just jump ahead to this figure in the same way as any other figure.

Fig 5C, D - label x-axis.

Added.

Fig 5E - axis labels needed. I don't know what is plotted on x and y, and cannot see red and green lines in left plot

Thanks for noticing this, colors corrected, axes labeled.

Page 8 L3-15 - If I follow this correctly, I think the authors are confusing pitch and frequency here in a way that is fundamental to their model. They seem to equate tonotopic frequency tuning to pitch tuning, leading to confused implications of frequency adaptation on the F0 representation of complex sounds like Shepard tones. To my knowledge, the authors do not examine pure tone frequency tuning in their neurons in this study. Please clarify how you propose that frequency tuning like that shown in Fig 5A relates to representation of the F0 of Shepard tones. Or...are the authors suggesting these neural effects have little to do with pitch processing and instead are just the result of frequency tuning for a single harmonic of the Shepard tones?

We agree that it is not trivial to describe this well, while keeping the text uncluttered, in particular, because often tuning properties to stimulus frequency contribute to tuning properties of the same neuron for pitch class, although this can be more or less straightforward: specifically, for some narrowly tuned cells, the Shepard tuning is simply a reflection of their tuning to a single octave range of the constituent tones (see Fig. S1). For more broadly tuned cells, multiple constituent tones will contribute to the overall Shepard tuning, which can be additive, subtractive, or more complex. The assumption in our approach is that we can directly estimate the Shepard tuning to evaluate the consequence for the percept. While this may seem artificial, as Shepard tones do not typically occur in nature, the same argument could be made against pure tones, on which classical tuning curves and associated decodings are often based. Relating the Shepard tuning to the classical tuning would be an interesting study in itself, although arguably relating the tuning of one artificial stimulus to another. Regarding the terminology of pitch, pitch class and frequency: The term pitch class is commonly used in the field of Shepard tones, and - as we indicated in the beginning of the results: "the term *pitch* is used interchangeably with *pitch class* as only Shepard tones are considered in this study". We agree that the term pitch, which describes the perceptual convergence/construction of a tone-height from a range of possible physical stimuli, needs to be separated from frequency as one contributor/basis for the perception of a pitch. However, we think that the term pitch can - despite its perceptual origin - also be associated with neuron/neural responses, in order to investigate the neural origin of the pitch percept. At the same time, the present study is not targeted to study pitch encoding per se, as this would require the use of a variety of stimuli leading to consistent pitch percepts. Therefore, pitch (class) is here mainly used as a term to describe the neural responses to Shepard tones, based on the previous literature, and the fact that Shepard tones are composite stimuli that lead to a pitch percept. The last sentence has been added to the manuscript for clarity.

P7-9: I wasn't left with a clear idea of how the model works from this text. I assume you have layers of neurons tuned to frequency or f0 (based on the real data?), which are connected in some way to produce some sort of output when you input a sound? More detail is needed here. How is the dynamic adaptation implemented?

The detailed description of the model can be found in the Methods section. We have gone through the corresponding paragraph and have tried to clarify the description of the model by introducing a high-level description and the reference to the corresponding Figure (Fig. 5A) in the Results.

Fig6A: Figure caption can't be correct. In any case, these equations cannot be understood unless you define the terms in them.

We have clarified the description in the caption.

Fig 6/directionality analysis: Assuming that the "F" in the STRFs here is Shepard tone f0, and not simple frequency?

We have changed the formula in the caption and the axis labels now.

Fig 6C - y-axis values

In the submission, these values were left out on purpose, as the result has an arbitrary scale, but only whether it is larger or smaller than 0 counts for the evaluation of the decoded directionality (at the current level of granularity). An interesting refinement would be to relate the decoded values to animal performance. We have now scaled the values arbitrarily to fit within [-1,1], but we would like to emphasize that only their relative scale matters here, not their absolute scale.

Fig 6E - can't both be abscissa (caption). I might be missing something here, but I don't see the "two stripes" in the data that are described in the caption.

Thank you. The typo is fixed. The stripes are most clearly visible in the right panel of Fig. 6E, red and blue, diagonally from top left to bottom right.

Fig 6G -I have no idea what this figure is illustrating.

This panel is described in the text as follows: "The resulting distribution of activities in their relation to the Bias is, hence, symmetric around the Bias (Fig. 6G). Without prior stimulation, the population of cells is unadapted and thus exhibits balanced activity in response to a stimulus. After a sequence of stimuli, the population is partially adapted (Fig. 6G right), such that a subsequent stimulus now elicits an imbalanced activity. Translated concretely to the present paradigm, the Bias will locally adapt cells. The degree of adaptation will be stronger, if their tuning curve overlaps more with the biased region. Adaptation in this region should therefore most strongly influence a cell’s response. For example, if one considers two directional cells, an up- and a down-selective cell, cocentered in the same frequency location below the Bias, then the Bias will more strongly adapt the up-cell, which has its dominant, recent part of the SSTRF more inside the region of the Bias (Fig. 6G right). Consistent with the percept, this imbalance predicts the tone to be perceived as a descending step relative to the Bias. Conversely, for the second stimulus in the pair, located above the Bias, the down-selective cells will be more adapted, thus predicting an ascending step relative to the previous tone."

I might be just confused or losing steam at this point, but I do not follow what has been done or the results in Fig 6 and the accompanying text very well at all. Can this be explained more clearly? Perhaps the authors could show spike rate responses of an example up-direction and down-direction neuron? Explain how the decoder works, not just the results of it.

We agree that we are presenting something new here. However, it is conceptually not very different from decoding based on preferred frequencies. We have attempted to provide two illustrations of how the decoder works (Fig. 6A) and how it then leads to the percept using prototypical examples of cellular SSTRFs (Fig. 6G). We have added a complete, but accessible description to the Methods section. Showing firing rates of neurons would unfortunately not be very telling, given the usual variability in neural response and the fact that our paradigm did not have a lot of repetitions (but instead a lot of conditions), which would be able to average out the variability on a single neuron level.

Discussion - I do not feel I can adequately critique the author's interpretation of the results until I understand their results and methods better. I will therefore save my critique of the discussion section for the next round of revisions after they have addressed the above issues of disorganization and clarity in the manuscript.

We hope that the updated version of the manuscript provides the reviewer now with this possibility.

MethodsP15L7 - gender of human subjects? Age distribution? Age of ferrets?

We have added this information.

P16L21 - What is the justification for randomizing the phase of the constituent frequencies?

The purpose of the randomization was to prevent idiosyncratic phase relationships for particular Shepard tones, which would depend in an orderly fashion on the included base-frequencies if non-randomized, and could have contributed to shaping the percept for each Shepard tone in a way that was only partly determined by the pitch class of the Shepard tone. Added to the section.

P17L6 - what are the 2 randomizations? What is being randomized?

Pitch classes and position in the Bias sequence. Added to the section.

P16 Shepard Tuning section - What were the durations of the tones and the time between tones within a trial?

Thanks, added!

Equations - several undefined terms in the equations throughout the manuscript.

Thanks. We have gone through the manuscript and all equations and have introduced additional definitions where they had been missing.

**Reviewer #3 (Recommendations For The Authors):**
P3L10: "passive" and "active" conditions come totally out of the blue. Need introducing first. (Or cut. If adaptation is always seen, why mention the two conditions if the difference is not relevant here?)

We have added an additional sentence in the preceding paragraph, that should clarify this. The reason for mentioning it is that otherwise a possible counter-argument could be made that adaptation does not occur in the active condition, which was not tested in ferrets (but presents an interesting avenue for future research).

P3L14 "siple" typo

Corrected.

P4L1 "behaving humans" you should elaborate just a little here on what sort of behavior the participants engaged in.

Thanks for pointing this out. We have clarified this by adding an additional sentence directly thereafter.

P4 adaptation: I wonder whether it would be useful to describe the Bias condition a bit more here before going into the observations. The reader cannot know what to expect unless they jump ahead to get a sense of what the Bias looks like in the sense of how many stimuli are in it, and how similar they are to each other. Observations such as "the average response strength decreases as a function of the position in the Bias sequence" are entirely expected if the Bias is made up of highly repetitive material, but less expected if it is not. I appreciate that it can be awkward to have Methods after Results, but with a format like that, the broad brushstroke Methods should really be incorporated into the Results and only the tedious details should be reserved for the Methods to avoid readers having to jump back and forth.

Agreed, we have inserted a corresponding description before going into the details of the results.

Related to this (perhaps): Bottom of P4, top of P5: "significantly less reduced (33%, p=0.0011, 2 group t-test) compared to within the bias (Fig. 2 A3, blue vs. red), relative to the first responses of the bias" ... I am at a loss as to what the red and blue symbols in Fig 2 A3 really show, and I wonder whether the "at the edges" to "within the Bias" comparison were to make sense if at this stage I had been told more about the composition of the Bias sequence. Do the ambiguous ('target') tones also occur within the Bias? As I am unclear about what is compared against what I am also not sure how sound that comparison is.

We have added an extended description of the Bias to the beginning of this section of the manuscript. For your reference: the Shepard tones that made up the ambiguous tones were not part of the Bias sequence, as they are located at 3st distance from the center of the Bias (above and below), while the Bias has a range of only +/- 2.5st.

Fig 2: A4 B1 B2 labels should be B1 B2 B3

Corrected.

Fig 2 A2, A3: consider adjusting y-axis range to have less empty space above the data. In A3 in particular, the "interesting bit" is quite compressed.

Done, however, while still matching the axes of A2 and A3 for better comparability.

I am under the strong impression that the human data only made it into Fig 2 and that the data from Fig 3 onwards are animal data only. That is of course fine MEG may not give responses that are differentiated enough to perform the sort of analyses shown in the later figures. But I do think that somewhere this should be explicitly stated.

Yes, the reviewer's observation is correct. The decoding analyses could not be conducted on the human MEG data and was therefore not further pursued. Its inclusion in the paper has the purpose of demonstrating that even in humans and active conditions, the local adaptation is present, which is a key contributor to the two decoding models. We now state this explicitly when starting the decoding analysis.

P5L2 "bias" not capitalized. Be consistent.

All changed to capitalized.

P5L8 reference to Fig 2 A4: something is amiss here. From legend of Fig 2 it seems clear that panel A4 label is mislabeled B1. Maybe some panels are missing to show recovery rates?

Apologies for this residual text from a previous version of the manuscript. We have gone through all references and corrected them.

P6L7 comma after "decoding".

Changed.

Fig 3, I like this analysis. What would be useful / needed here though is a little bit more information about how the data were preprocessed and pooled over animals. Did you do the PCA separately for each animal, then combine, or pool all units into a big matrix that went into the PCA? What about repeat, presentations? Was every trial a row in the matrix, or was there some averaging over repeats? (In fact, were there repeats??)

Thanks for bringing up these relevant aspects, which were partly insufficiently detailed in the manuscript. Briefly, cells were pooled across animals and we only used cells that could meaningfully contribute to the decoding analysis, i.e. had auditory responses and different responses to different Shepard tones. Regarding the responses, as stated in the Methods, "Each stimulus was repeated 10 times", and we computed average responses across these repetitions. Single trials were not analyzed separately. We have added this information in the Methods, and refer to it in the Results.

Also, there doesn't appear to be a preselection of units. We would not necessarily expect all cortical neurons to have a meaningful "best pitch" as they may be coding for things other than pitch. Intuitively I suspect that, perhaps, the PCA may take care of that by simply not assigning much weight to units that don't contribute much to explained variance? In any event I think it should be possible, and would be of some interest, to pull out of this dataset some descriptive statistics on what proportion of units actually "care about pitch" in that they have a lot (or at least significantly more than zero) of response variance explained by pitch. Would it make sense to show a distribution of %VE by pitch? Would it make sense to only perform the analysis in Fig 3 on units that meet some criterion? Doing so is unlikely to change the conclusion, but I think it may be useful for other scientists who may want to build on this work to get a sense of how much VE_pitch to expect.

We fully agree with the reviewer, which is why this information is already presented in Supplementary Fig.1, which details the tuning properties of the recorded neurons. Overall, we recorded from 1467 neurons across all ferrets, out of which 662 were selected for the decoding analysis based on their driven firing rate (i.e. whether they responded significantly to auditory stimulation) and whether they showed a differential response to different Shepard tones The thresholds for auditory response and tuning to Shepard tones were not very critical: setting the threshold low, led to quantitatively the same result, however, with more noise. Setting the thresholds very high, reduced the set of cells included in the analysis, and eventually that made the results less stable, as the cells did not cover the entire range of preferences to Shepard tones. We agree that the PCA based preprocessing would also automatically exclude many of the cells that were already excluded with the more concrete criteria beforehand. We have added further information on this issue in the Methods section under the heading 'Unit selection'.

P9 "tones This" missing period.

Changed.

P10L17 comma after "analysis"

Changed.